# Caveolae and Bin1 form ring-shaped platforms for T-tubule initiation

Eline Lemerle[1], Jeanne Lainé[1,2], Marion Benoist[1], Gilles Moulay[1], Anne Bigot[1], Clémence Labasse[3], Angéline Madelaine[3], Alexis Canette[4], Perrine Aubin[5], Jean-Michel Vallat[6], Norma B Romero[1,3], Marc Bitoun[1], Vincent Mouly[1], Isabelle Marty[5], Bruno Cadot[1], Laura Picas[7], Stéphane Vassilopoulos[1]*

[1]Institut de Myologie, Sorbonne Université, INSERM, Paris, France; [2]Department of Physiology, Faculty of Medicine Pitié-Salpêtrière, Sorbonne Université, Paris, France; [3]Neuromuscular Morphology Unit, Institut de Myologie, Pitié-Salpêtrière Hospital, Sorbonne Université, Paris, France; [4]Sorbonne Université, CNRS, Institut de Biologie Paris-Seine (IBPS), Service de Microscopie Électronique (IBPS-SME), Paris, France; [5]Université Grenoble Alpes, INSERM, U1216, Grenoble Institut des Neurosciences, Grenoble, France; [6]Department of Neurology, National Reference Center for 'Rare Peripheral Neuropathies', University Hospital, Limoges, France; [7]Institut de Recherche en Infectiologie de Montpellier, CNRS UMR 9004, Université de Montpellier, Montpellier, France

*For correspondence:
s.vassilopoulos@institut-myologie.org

**Abstract** Excitation-contraction coupling requires a highly specialized membrane structure, the triad, composed of a plasma membrane invagination, the T-tubule, surrounded by two sarcoplasmic reticulum terminal cisternae. Although the precise mechanisms governing T-tubule biogenesis and triad formation remain largely unknown, studies have shown that caveolae participate in T-tubule formation and mutations of several of their constituents induce muscle weakness and myopathies. Here, we demonstrate that, at the plasma membrane, Bin1 and caveolae composed of caveolin-3 assemble into ring-like structures from which emerge tubes enriched in the dihydropyridine receptor. Bin1 expression lead to the formation of both rings and tubes and we show that Bin1 forms scaffolds on which caveolae accumulate to form the initial T-tubule. Cav3 deficiency caused by either gene silencing or pathogenic mutations results in defective ring formation and perturbed Bin1-mediated tubulation that may explain defective T-tubule organization in mature muscles. Our results uncover new pathophysiological mechanisms that may prove relevant to myopathies caused by Cav3 or Bin1 dysfunction.

## Editor's evaluation

Lemerle et al. provide convincing evidence that advances our fundamental understanding of how transverse tubules may be formed, a significant gap in our understanding of excitation contraction coupling and muscle biology more broadly. They utilize advanced correlative light and electron microscopy and molecular biology approaches to demonstrate the presence of Bin1 and caveolae containing rings that are capable and necessary to properly tubulate membranes in developing striated muscle.

## Introduction

Muscle contraction is induced by the release of intracellular calcium resulting from the transmission of the neuronal action potential to the muscle fiber. This step called excitation-contraction coupling

(E-C coupling), relies on the physical interaction between the dihydropyridine receptor (DHPR) and the ryanodine receptor type 1 (RyR1), two calcium channels anchored in different membrane compartments of the muscle cell (*Franzini-Armstrong, 2018*; *Marty et al., 1994*). The DHPR is a voltage-activated calcium channel present in invaginations of the plasma membrane called transverse tubules (T-tubules) and RyR1 is the main calcium channel located in cisternae of the sarcoplasmic reticulum (SR) in close proximity to T-tubules (*Franzini-Armstrong and Protasi, 1997*). The function of this calcium release complex relies on a unique membrane system composed of two SR cisternae apposed to one T-tubule, a structure named the triad. While the structure and function of the triad are now well characterized, its biogenesis remains poorly understood, especially regarding the formation of the T-tubule.

It was initially thought that T-tubule formation occurs in two stages (*Flucher et al., 1991*). In the first step, tubules emanate from the plasma membrane longitudinally. In the second step, tubules subsequently meet and become connected with a cytosolic membrane compartment. An involvement of caveolae has long been suspected in the formation of the nascent T-tubules from mouse, chick, and rat skeletal muscle (*Franzini-Armstrong, 1991*; *Ishikawa, 1968*; *Schiaffino et al., 1977*). Caveolae are small cave-like indentations of the plasma membrane that are very abundant in skeletal muscle, where they have many functions including signal transduction, lipid homeostasis and mechanoprotection (*Parton and Simons, 2007*). Caveolins were the first caveolae-constituting molecules described; caveolin-1 (Cav1) and caveolin-3 (Cav3) but not caveolin-2 are required for caveolae formation at the membrane of non-muscle and muscle cells (*Capozza et al., 2005*; *Fra et al., 1995*; *Rothberg et al., 1992*). Caveolins associate with cavin family proteins (cavins 1–4) to stabilize caveolae at the plasma membrane. Cavin 1 (also known as Polymerase I and transcript release factor PTRF) is essential for caveolae formation in mammalian cells (*Hill et al., 2008*; *Liu et al., 2008*) by interacting with Cav1 (in non-muscle cells) and Cav3 (in muscle cells). The muscle-specific cavin 4 isoform (also known as muscle-restricted coiled-coil, MURC), the last described caveolae protein (*Ogata et al., 2008*), is not required for caveolae formation but may play a role in caveolar morphology (*Bastiani et al., 2009*; *Lo et al., 2015*). In skeletal muscle, Cav3 localizes mostly on the plasma membrane of mature fibers but is also present at the core of the fiber where it is associated with the T-tubule system (*Ralston and Ploug, 1999*). Plasma membrane openings of caveolae and T-tubules are of the same size (20–40 nm) and appear indistinguishable by electron microscopy (EM) (*Franzini-Armstron, 1974*; *Franzini-armstrong and Porter, 1964*). This similarity between caveolae and T-tubules (*Ishikawa, 1968*), as well as their dependence on membrane cholesterol (*Hidalgo et al., 1986*; *Rothberg et al., 1992*; *Yuan et al., 1991*), the enrichment of caveolae at the neck of T-tubules at the plasma membrane (*Franzini-Armstrong, 1991*; *Murphy et al., 2009*), and the association of Cav3 with developing T-tubules (*Parton et al., 1997*) all suggest that caveolae and their associated proteins could have a role in T-tubule biogenesis. It has been suggested that caveolae may fuse together and form a substrate for proteins capable of tubulating membranes such as Bin1 (also known as amphiphysin 2) (*Lee et al., 2002*) and more recently, a direct interaction between Bin1 and cavin 4 was shown to be required for normal T-tubule formation in zebrafish muscle (*Lo et al., 2021*). The importance of these proteins in skeletal muscle physiology is emphasized by the fact that mutations in the *CAV3* gene cause autosomal dominant neuromuscular diseases called caveolinopathies (*Aboumousa et al., 2008*; *Gazzerro et al., 2010*; *Woodman et al., 2004*) and mutations in *BIN1* induce T-tubule and triad defects in the autosomal recessive form of centronuclear myopathy (*Nicot et al., 2007*). In addition, Bin1 splicing defects in the exon responsible for the interaction of Bin1 with membrane phosphoinositides induce T-tubule defects in myotonic dystrophy (*Fugier et al., 2011*). In patients with caveolae deficiency and in *Cav3* knock-out mice, defects in T-tubule orientation and defects in calcium flow are observed, although T-tubules still develop (*Galbiati et al., 2001*). Despite its major importance, the involvement of caveolins in T-tubule formation and the effect of caveolin mutations on the T-tubule system remain poorly understood.

To explore the role of caveolae in T-tubule formation, we examined the nanoscale composition of nascent T-tubules by a correlative microscopy technique combining super-resolution fluorescence and electron microscopy on platinum replicas (PREM) applied to control and genetically modified human and murine skeletal muscle myoblasts differentiated into multinucleated myotubes. We discovered the existence of novel structures that we termed caveolae rings, formed by Cav3 and Bin1. We found that caveolae rings are formed on Bin1 scaffolds that create contact sites with RyR1-positive ER/SR

tubes, are enriched in DHPR and act as platform for T-tubule nucleation. In addition, we show that the Bin1-induced membrane tubulation requires proper Cav3 function. Collectively, this work suggests that caveolae rings are the long-awaited sites for T-tubule initiation and elongation and provides the basis to better understand T-tubule biogenesis in healthy skeletal muscle and its defect in the pathophysiology of caveolinopathies.

## Results

### Cav3 caveolae form nanoscale rings in differentiated myotubes

Nascent T-tubules initially have a longitudinal orientation during in vitro differentiation, and progressively gain a transversal orientation reflecting extensive differentiation (*Flucher et al., 1993*; *Takekura et al., 2001*). To study the formation of T-tubules during myogenic differentiation, we embedded primary murine myotubes in two layers of an extracellular matrix hydrogel (Matrigel) to avoid detachment by spontaneous contractions and supplemented the differentiation medium with agrin to promote the progression of myotube maturation toward contracting myofibers (*Falcone et al., 2014*). When myotubes produced from murine primary myoblasts were differentiated for 10 days, the characteristic labeling of mature triads in double transverse rows of dots was clearly observed while at 5 days, DHPR and RyR1 still presented a longitudinal distribution in clusters (*Figure 1—figure supplement 1*). To visualize the interplay between caveolae and nascent T-tubules at the surface of well-differentiated myotubes, we unroofed myotubes using ultrasounds to access the inner side of the plasma membrane for both light and electron microscopy analyses. Unroofing allowed us to remove the intracellular structures whose labeling could hinder identification of structures at the plasma membrane and to avoid permeabilization with detergents. At the light microscopy level, tagging Cav3 with GFP produced a very characteristic pattern on unroofed myotubes with numerous circular or crescent-shaped structures from which tubules emanated (*Figure 1A*, insets 1–5). These structures were observed on both unroofed human myotubes expressing constitutively Cav3$^{GFP}$ and unroofed primary mouse myotubes labeled with antibodies against Cav3 (*Figure 1B*). The diameter of these rings from either human myotubes expressing Cav3$^{GFP}$ or murine myotubes labeled with antibodies against Cav3 was the same, 652±174 nm and 637±183 nm, respectively. To compare the localization of an SR marker with respect to Cav3, we labeled both Cav3 and the intracellular RyR1 Ca$^{2+}$ channel. The two proteins were partially colocalized but marked two clearly distinct networks. RyR1-labeled structures were often surrounded by or in close proximity to Cav3 rings (insets in *Figure 1A*). We next produced platinum replicas of unroofed human and mouse primary myotubes for electron microscopy (EM) analysis. At the ultrastructural level, in addition to the numerous individual caveolae decorating the cytoplasmic side of the plasma membrane, we confirmed the presence of circular assemblies composed of bona fide caveolae in human myotubes (*Figure 1C–E*). These 'caveolae rings' were composed of assembled caveolae forming a circle. We also found instances of rings where caveolae would accumulate at one pole of the circle (*Figure 1D*) and caveolae rings that were intertwined with cortical SR (*Figure 1E*). The central part of the ring contained globular membrane proteins (*Figure 1C–E*, orange circles). We observed similar caveolae rings in mouse primary myotubes using both PREM and conventional thin-section EM (*Figure 1*, F-G). From measurements on electron micrographs, we determined an average diameter for these rings of 573 nm in human myotubes and 546 nm in murine primary myotubes (*Figure 1H*). We counted 16.9 and 16.1 caveolae per ring on average in human and mouse myotubes respectively (*Figure 1I*). Caveolae from primary mouse myotubes were significantly smaller than their human counterparts (64 *vs* 85.4 nm, respectively) (*Figure 1J*). In order to simultaneously visualize the ultrastructure of myotubes and the localization of proteins of interest, we developed a correlative light and electron microscopy (CLEM) assay combining observation of Cav3$^{GFP}$ by super-resolution fluorescent microscopy with the ultrastructure of platinum replicas observed by electron microscopy. Using this CLEM approach, we were able to show that the circular Cav3 assemblies seen at the light microscopy level correspond to rings formed by caveolae (*Figure 1*, K-M).

In order to label the nascent tubes we used antibodies against Bin1, a bona fide T-tubule component whose expression strongly increases during differentiation and correlates with increased expression of Cav3 (*Butler et al., 1997*; *Lee et al., 2002*). We compared the localization of Bin1 with the localization of the triad protein marker RyR1 and observed Bin1 labeling on a network of predominantly longitudinal T-tubules and, as expected, a partial colocalization of these tubules with RyR1-positive

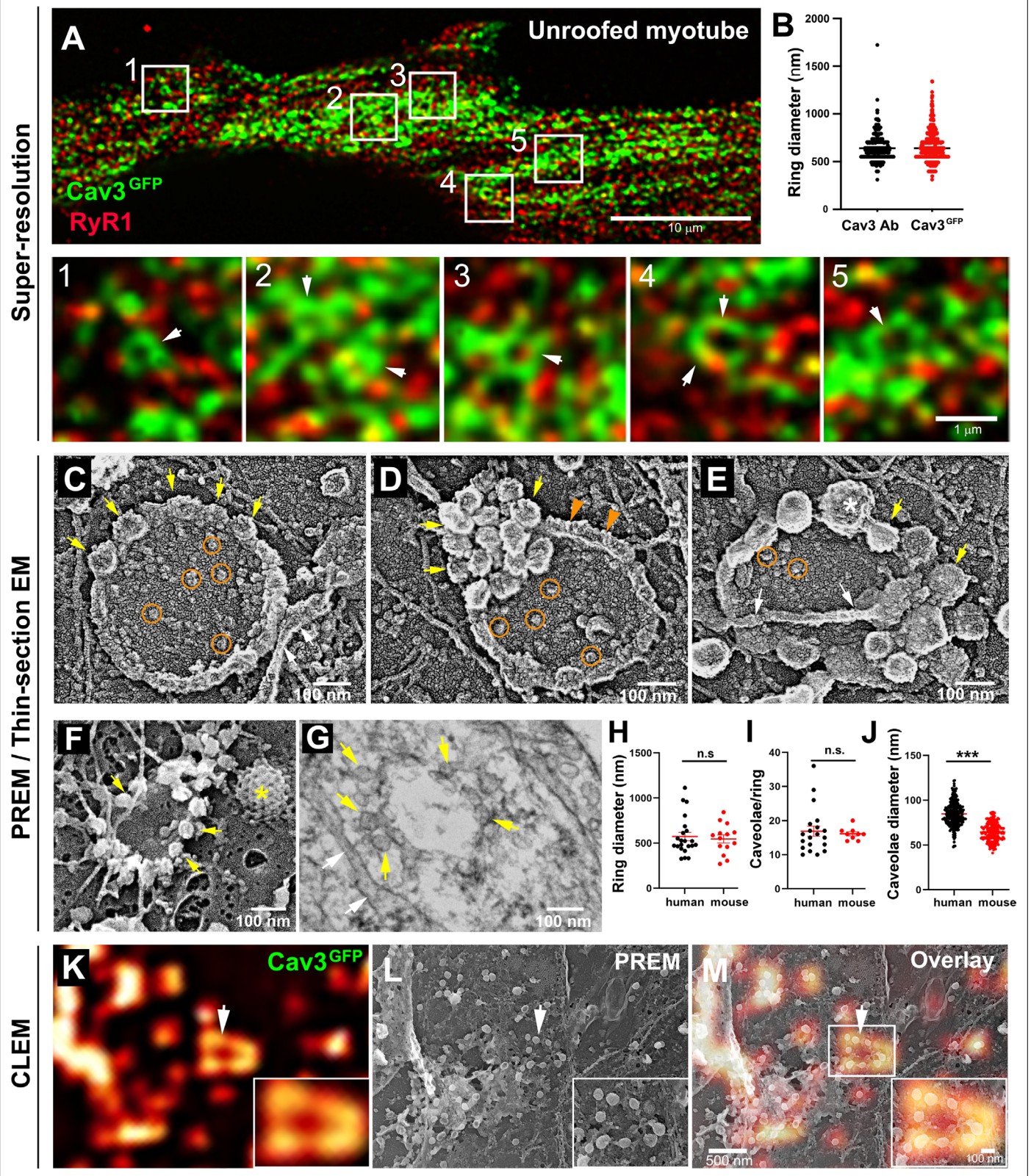

**Figure 1.** Cav3-positive caveolae form nanoscale rings in differentiated myotubes. (**A**) Super-resolution images of a human myotube expressing Cav3<sup>GFP</sup> and labelled with antibodies against RyR1 (red). (**B**) Quantification of ring diameter from super-resolution images of differentiated myotubes either expressing Cav3<sup>GFP</sup> or labelled with antibodies against Cav3 (Cav3Ab, n=21 myotubes from three independent experiments; Cav3<sup>GFP</sup>, n=14 myotubes from three independent experiments; p=0.081). (**C–E**) Gallery of PREM images of rings in unroofed human myotubes. Yellow arrows indicate individual

*Figure 1 continued on next page*

Figure 1 continued

caveolae found on caveolae rings, white arrows denote cortical ER/SR cisternae and orange circles denote protein particles present inside the ring. (F) High magnification view of a ring formed by groups of caveolae in unroofed mouse primary myotubes. Yellow arrows denote caveolae. The yellow asterisk indicates a clathrin-coated pit. (G) High-magnification view of a ring formed by caveolae in unroofed mouse primary myotubes observed on thin-sections. Yellow arrows indicate groups of caveolae with a circular organization and white arrows denote cortical ER/SR. (H) Quantification of ring diameters from human and mouse myotubes on PREM images (human, n=22 images from four independent experiments; mouse, n=14 images from five independent experiments; p=0.67). (I) Quantification of the number of caveolae composing a ring from differentiated myotubes (human, n=20; mouse, n=9; p=0.63). (J) Quantification of the average caveolae diameter from differentiated mouse or human myotubes (human, 439 caveolae from n=15 images; mouse, 187 caveolae from n=12 images; p<0.0001). Quantification was performed from at least three independent experiments. (K–M) CLEM images combining Cav3$^{GFP}$ super-resolution microscopy with PREM of the same human myotube. (K) Cav3$^{GFP}$ fluorescence. (L) Same area imaged with TEM. (M) CLEM overlay. White arrow indicates a group of caveolae with a circular organization.

The online version of this article includes the following video, source data, and figure supplement(s) for figure 1:

**Source data 1.** Quantification of ring diameter in Cav3$^{GFP}$ and immunolabeled myotubes.

**Source data 2.** Quantification of ring diameter from human and mouse myotubes on PREM.

**Source data 3.** Quantification of the number of caveolae per ring.

**Source data 4.** Quantification of the average caveolae diameter.

**Figure supplement 1.** Triad markers are organized longitudinally and then transversally during in vitro differentiation of myotubes into myofibers.

**Figure supplement 2.** Nascent T-tubules form ring-like structures in myotubes.

**Figure supplement 2—source data 1.** Quantification of ring diameter labeled by MemBright.

**Figure 1—video 1.** T-tubule staining on live myotubes corresponding to *Figure 1—figure supplement 2*.
https://elifesciences.org/articles/84139/figures#fig1video1

SR membranes (*Figure 1—figure supplement 2*). In order to label lipids and trace nascent T-tubules without relying on a protein marker, we used the MemBright fluorescent lipid probe (*Collot et al., 2019*). This cell-impermeable probe renders biological membranes fluorescent and allows the visualization of the intracellular tubular network emanating from the plasma membrane. We incubated myotubes with MemBright and acquired optical stacks (0.1–0.3 µm step) through the entire myotube with a spinning-disk confocal microscope equipped with a structured illumination module allowing sub-diffraction light microscopy (*Figure 1—figure supplement 2* and *Figure 1—video 1*). After a 10 min incubation on live myotubes, the probe had diffused in the tubule in the depth of the fiber and labeled a tubular membrane system, continuous with the plasmalemma. Interestingly, the dye labeled circular structures on both the dorsal and ventral surface of myotubes, presenting an average diameter of 667±286 nm (*Figure 1—figure supplement 2*). Analysis of the whole fiber volume confirmed that these tubules were more abundant near the surface although a few tubes penetrated deep into the core of the muscle cell producing numerous bifurcations (*Figure 1—figure supplement 2* and *Figure 1—video 1*). This tubular system was identical to the one described in previous attempts to label the T-tubule system in cultured cells and muscle fibers with DiIC$_{16}$ (*Flucher et al., 1993*). At the ultrastructural level, thin-section EM captured numerous pearled tubules forming contacts with SR membranes presenting the characteristic RyR1 electron densities (*Franzini-Armstrong, 2018*; *Lainé et al., 2018*, *Figure 1—figure supplement 2*). We noticed the presence of small 60–70 nm buds on discrete regions of forming T-tubules with characteristics of caveolae. Frequent electron dense couplings between SR membrane and tubules formed on parts of the tubule were devoid of these caveolae-like buds.

## Cav3 rings and tubules form contacts with SR terminal cisternae

While analyzing caveolae rings by CLEM, we observed a proximity with the cortical endo/sarcoplasmic reticulum (*Figure 2A*). We found numerous instances where the cortical ER/SR either contacted the center or was entangled with the caveolae ring (*Figure 2B–C*). We never observed direct SR-caveolae contacts but rather an interaction between the SR and regions of the ring devoid of caveolae, confirming our previous observations using super-resolution (*Figure 1A*). To demonstrate the ER/SR nature of these membrane tubules, we performed CLEM analysis by double labeling Cav3 and the intracellular RyR1 Ca$^{2+}$ channel (*Figure 2D–E* and *Figure 2—video 1*). The two proteins labeled two distinct but overlapping networks with Cav3 labeling individual caveolae, rings and tubules while RyR1 labeled the cortical SR cisternae surrounding them.

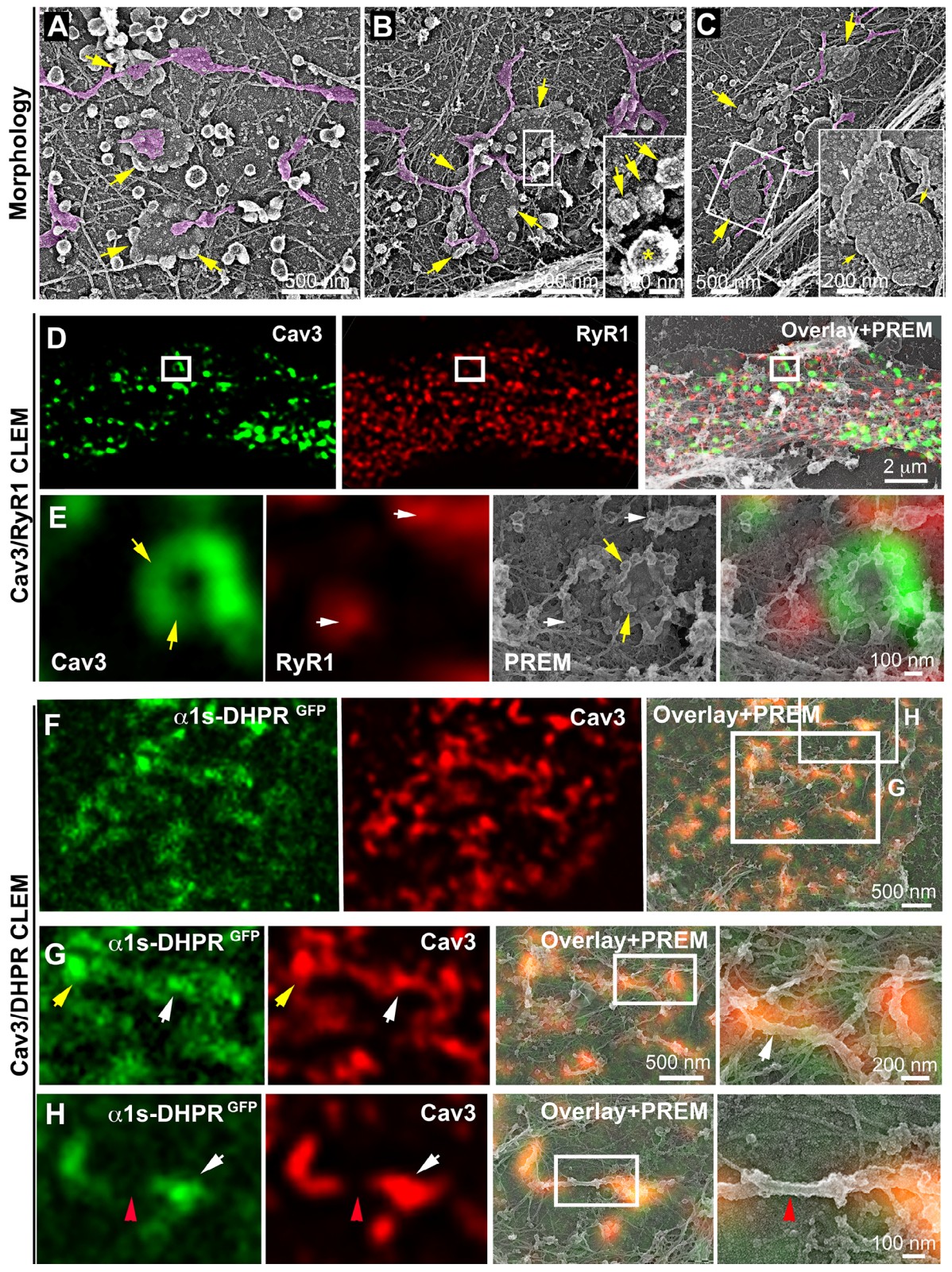

**Figure 2.** DHPR is enriched in Cav3 structures in contact with RyR1-positive SR cisternae. (**A–C**) EM images showing caveolae rings in contact with cortical endo/sarcoplasmic reticulum cisternae (pseudo-colored purple) on the cytosolic part of the adherent sarcolemma. Yellow arrows indicate caveolae rings. Caveolae rings are formed by caveolae still presenting their characteristic coat indicated by yellow arrows in B inset (yellow asterisk indicates a clathrin-coated pit). White arrows indicate endo/sarcoplasmic reticulum cisternae going over the caveolae ring. (**D–E**) Correlative microscopy

*Figure 2 continued on next page*

*Figure 2 continued*

of Cav3 (green) and RyR1 (red) labeling on myotubes. (**E**) Enlargement of the inset in (**D**). Yellow arrows indicate caveolae rings. White arrows indicate cortical endo/sarcoplasmic reticulum cisternae. (**F–H**) Correlative microscopy of the α1s-subunit of DHPR fused to GFP (green) and Cav3 (red) labeling on myotubes. Yellow arrows indicate a caveolae ring. White arrows indicate tubules where Cav3 and DHPR colocalize. Red arrowheads indicate a region of the tubule free of Cav3 and α1s labeling and lacking caveolar material at the ultrastructural level.

The online version of this article includes the following video and figure supplement(s) for figure 2:

**Figure supplement 1.** Additional CLEM of DHPR in caveolin-positive ring-like structures.

**Figure 2—video 1.** Correlative super-resolution/PREM for Cav3 and RyR1 corresponding to *Figure 2D*.

https://elifesciences.org/articles/84139/figures#fig2video1

## Caveolae rings extend into Bin1-positive tubules

We next wanted to know if rings could concentrate T-tubule proteins (i.e DHPR) at the plasma membrane and associate with the cortical SR to form early excitation-contraction coupling sites, allowing these two distinct membrane compartments to come into contact early during tubule biogenesis. To test this, we expressed the GFP-tagged version of the α1s-subunit of the DHPR using a lentivirus vector that we transduced prior to differentiating the myoblasts into myotubes. In agreement with previous work linking DHPR and Cav3 (*Couchoux et al., 2007*; *Weiss et al., 2008*) we showed co-localization of both the DHPRs α1s-subunit and Cav3 (*Figure 2F–G* and *Figure 2—figure supplement 1*). CLEM analysis confirmed that the α1s-subunit of the DHPR accumulated in regions of the plasma membrane enriched in Cav3-positive circular caveolae regions from which tubules appear. It is noteworthy that while both DHPR and Cav3 co-localized on nascent tubules, the labeling was only present on portions of the tubule containing the caveolae, suggesting a possible retention of the Ca²⁺ channel in caveolar regions (*Figure 2H*). The abundance of Cav3-positive circular structures during critical steps of T-tubule formation suggested a possible contribution of caveolae rings to their biogenesis. By unroofing the myotubes, we noticed that tubules with a characteristic beaded protein coat similar to the caveolar coat emerged from caveolae rings (*Figure 3A–C*). EM analysis of ultrathin sections from the adherent surface of myotubes confirmed the presence of *bona fide* caveolae rings from which numerous 25 nm diameter tubules emanated (*Figure 3D* and *Figure 3—figure supplement 1*). To validate this finding, we turned to CLEM and tested the hypothesis that caveolar rings could give rise to platforms for elongation of Bin1-positive tubules (*Figure 3E–L*). We performed double Cav3 and Bin1 immunolabeling before producing replicas of the same myotubes. Using our CLEM approach, we showed that endogenous Cav3 and Bin1 are present on caveolae rings but also on the tubules emanating from these rings (*Figure 3E–H and I–L*, *Figure 3—video 1* and *Figure 3—video 2*).

To further demonstrate that tubules extend from caveolae rings, we performed live imaging of myotubes stably expressing Cav3ᴳᶠᴾ. Numerous Cav3-positive rings could be resolved on the myotube surface. Our imaging captured numerous tubules emanating from rings and could measure tubule extension (*Figure 4A* and *Figure 4—video 1*, *Figure 4—video 2*). Analysis of their dynamics showed Cav3ᴳᶠᴾ-labeled structures moving at 45.7±22 nm/s. Interestingly, bright Cav3 spots appeared before the tubule at the same site where the tubule emanated (yellow arrows in *Figure 4A, B*) suggesting that Cav3 polarized accumulation observed by PREM (*Figure 1D, E*) could precede tubulation. We next analyzed Bin1 dynamics by expressing Bin1ᴳᶠᴾ in differentiated human myotubes (*Figure 4C, D* and *Figure 4—video 3*, *Figure 4—video 4*). As with Cav3ᴳᶠᴾ expression, Bin1 labeled both rings and tubules emanating from rings. Analysis of their dynamics showed that Bin1 labeled structures move at 64.2±18 nm/s on average. Thus, live-imaging confirms the CLEM evidence and provides the dynamics of tubules emanating from Cav3-positive rings.

## Bin1 is sufficient to form rings on the plasma membrane

We next focused on the role of Bin1 regarding its membrane curvature properties and its previous demonstrated links with T-tubules (*Fugier et al., 2011*; *Lee et al., 2002*; *Razzaq et al., 2001*). While all Bin1 isoforms electrostatically interact with negatively charged lipids through the inner surface of their BAR domain, only the isoform containing exon 11 specifically interacts with phosphatidylinositol 4,5-bisphosphate (PI4,5P₂). This exon is specifically included during skeletal muscle differentiation (*Fugier et al., 2011*; *Lee et al., 2002*; *Razzaq et al., 2001*). We used adenoviral vectors to overexpress either the Bin1ᴳᶠᴾ protein including exon 11 (Bin1ᴳᶠᴾ) or the ubiquitous isoform missing exon 11 (Bin1-Δex11ᴳᶠᴾ) (*Fugier et al., 2011*) to determine if the Bin1-mediated tubulation is associated

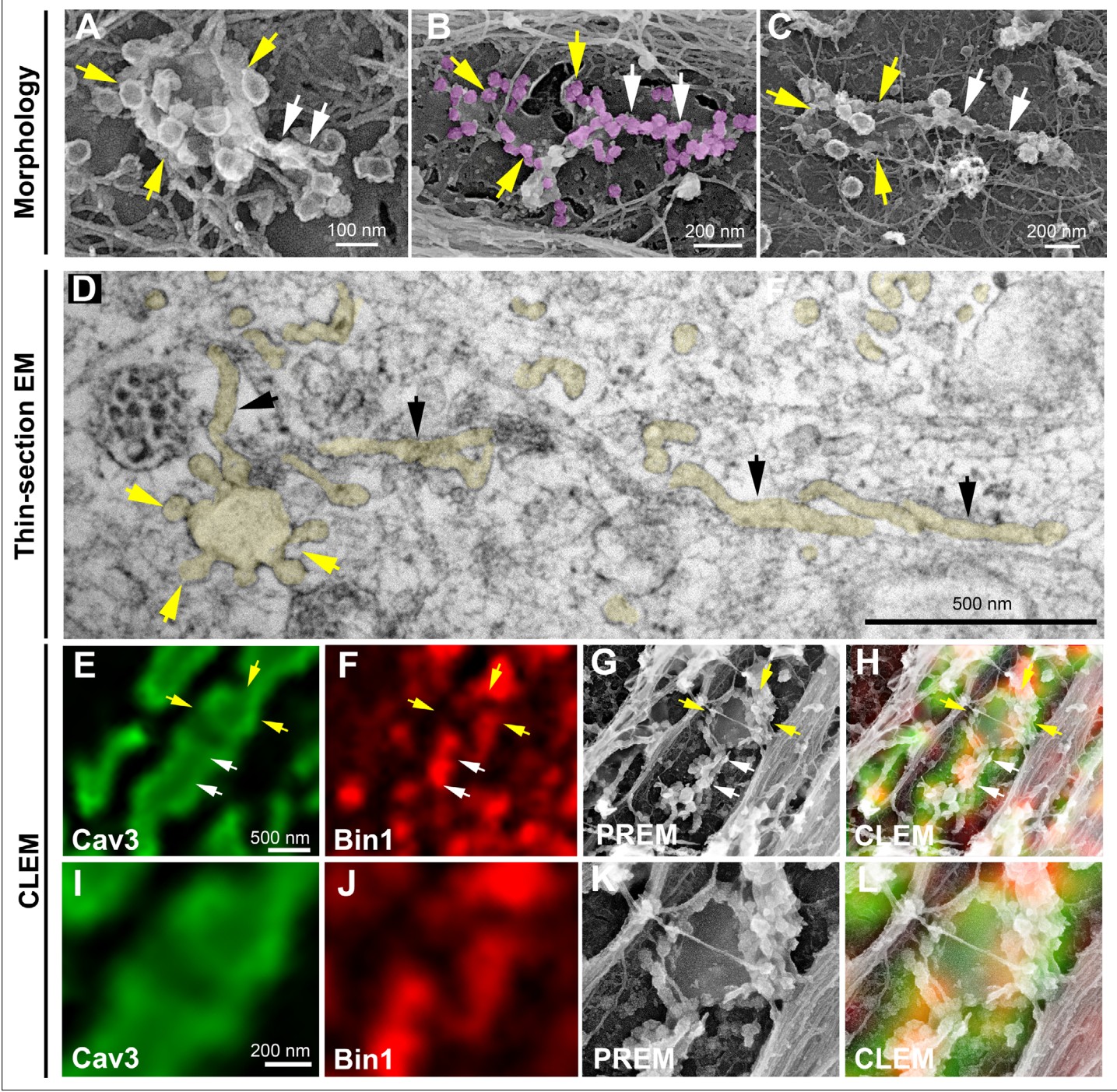

**Figure 3.** Caveolae rings extend into Bin1-positive tubules. (**A–C**) High magnification PREM views of caveolae rings on unroofed myotubes from primary mouse cultures. In (**B**) caveolae are pseudo-colored in light purple. White arrows indicate beaded tubes emanating from structures formed by ring caveolae and yellow arrows indicate the caveolae ring. (**D**) High magnification view of several tubules (black arrows) protruding from a single caveolae ring (yellow arrows) observed on conventional thin-section EM. The caveolae ring and tubules are pseudo-colored pale yellow. (**E–L**) Correlative microscopy of Cav3 (green) and Bin1 (red) immunolabeling on an unroofed 9 day extensively differentiated murine myotube. In (**E–H**), white arrows indicate tubules emanating from Bin1 and Cav3 labeled rings and yellow arrows indicate the ring formed by caveolae. (**I–L**) Higher magnification CLEM view of the ring structure labeled with Cav3 (green) and Bin1 (red) antibodies.

The online version of this article includes the following video and figure supplement(s) for figure 3:

**Figure supplement 1.** Thin-section EM analysis of extensively differentiated myotubes.

**Figure 3—video 1.** Correlative super-resolution/PREM of Cav3 and Bin1 corresponding to *Figure 3E–L*.

*Figure 3 continued on next page*

*Figure 3 continued*

https://elifesciences.org/articles/84139/figures#fig3video1

**Figure 3—video 2.** Correlative super-resolution/PREM of Cav3 and Bin1 corresponding to *Figure 3E-L*.
https://elifesciences.org/articles/84139/figures#fig3video2

with the formation of caveolar rings at the plasma membrane. We transduced human myotubes for 24 hr and showed that while both Bin1-Δex11$^{GFP}$ and Bin1$^{GFP}$-induced formation of caveolae rings with similar diameters and at the same density, only Bin1$^{GFP}$-induced formation of tubules emanating from the rings (*Figure 5—figure supplement 1*). Bin1$^{GFP}$ formed circular structures at the surface of myotubes that were also visible on unroofed myotubes (*Figure 5A* and *Figure 5—figure supplement 1*). Immunofluorescent labeling with antibodies against endogenous Cav3 confirmed that these rings were also present on intact myotubes (*Figure 5B*) and that Cav3 co-localized with Bin1 on rings and at the base of Bin1 tubules. Interestingly, while all the rings were positive for Bin1, Cav3 labeled only a subset suggesting that Bin1 acts upstream of Cav3 (*Figure 5B*). To visualize the structures formed by Bin1$^{GFP}$, PREM was performed on unroofed myotubes. Bin1 expression induced the formation of numerous beaded tubular structures that emanated radially from rings (*Figure 5C, D*). These structures were invariably composed of a ring from which a main pearled tubule and numerous smaller tubules emerged radially.

To understand the contribution of Bin1 to the formation of caveolae rings, we turned to in vitro experiments analyzing the impact of recombinant full-length Bin1 protein (isoform including exon 11) on supported lipid bilayers (SLBs) containing 5% of PI4,5P$_2$. PREM allowed us to directly visualize the membrane remodeling effect of recombinant Bin1 at 1 µM (*Figure 5E, F*). It has been described that proteins of the Amphiphysin family, at high protein concentration (i.e. ≥1 µM), can deform membranes by forming scaffold-like structures, as compared to lower protein concentrations where they mainly act as curvature sensors (*Sorre et al., 2012*). Along this line, we observed that at 1 µM, recombinant Bin1 assembled into circular structures from which tubules emerged radially. These structures, firmly bound to the lipid bilayer, resembled those formed on the surface of cultured myotubes indicating that Bin1 is able to spontaneously form these scaffolds on flat membranes both in vitro and in cellulo. Using the same in vitro assay combined to sub-diffraction light microscopy, we analyzed recombinant Bin1 self-assembly dynamics on SLBs. We showed that Bin1 forms stable ring-like structures and tubules that persist over several minutes (*Figure 5G* and kymographs in *Figure 5—figure supplement 2*). Importantly, we observed that Bin1 overexpression produced both pearled tubes with caveolar material and assemblies that were similar as those observed on SLBs with recombinant Bin1 (*Figure 5H*). The average diameter of Bin1 assemblies in vitro and in cellulo was 378±176 nm and 746±223 nm, respectively. The average length of the longest tube emanating from the central ring was 1356 nm in vitro and 2200 nm in cellulo while the average length of the smaller tubules emanating from the central ring was 420 nm in vitro and 590 nm in cellulo (*Figure 5I*).

## Bin1 forms Cav3-positive circular and tubular structures connected to the surface

These observations led us to analyze caveolae and Cav3 nanoscale organization at the sarcolemma of human and mouse myotubes expressing Bin1$^{GFP}$ by CLEM (*Figure 6*). Bin1 overexpression in human myotubes induced formation of pearled tubes with the same caveolar coat (*Figure 6A* and inset in 3D anaglyph). CLEM analysis with Bin1$^{GFP}$ allowed us to show that Bin1 labeled tubules formed in regions enriched with caveolae (*Figure 6B, D*). We observed that the formation of tubular and circular structures of the plasma membrane perfectly correlating with Bin1 fluorescence. However, the fluorescence corresponding to Bin1 localization was not always associated with caveolae rings at the ultrastructural level (*Figure 6E, G*). These circular structures, which resemble caveolae rings, suggested that Bin1 could self-assemble into platforms which act as scaffolds for recruitment of caveolae. We confirmed these observations by transducing primary mouse myotubes (*Figure 6H, K* and *Figure 6—video 1*). Bin1 overexpression lead to the formation of numerous Cav3-positive rings from which pearled tubules emerged (*Figure 6I, K*).

We next tested whether the tubules formed by Bin1 overexpression were connected to the surface. Caveolae openings (25–40 nm in diameter) can be visualized by observing the cells from the outside

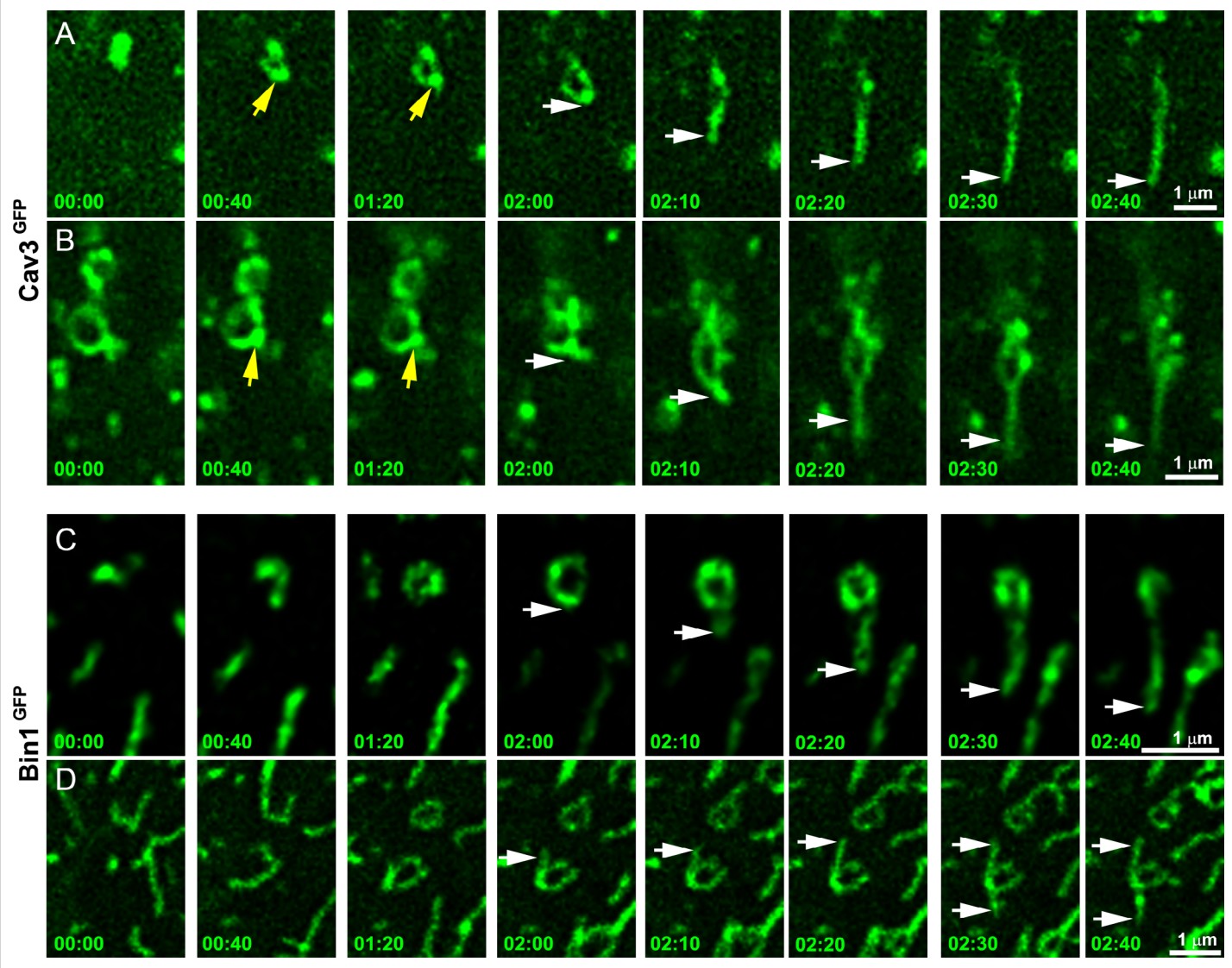

**Figure 4.** Time-lapse imaging of Cav3/Bin1 tubules extending from rings. (**A**) Gallery of consecutive frames from two time-lapse sequences of Cav3GFP expressing human myotubes (green). The gallery shows four consecutive frames every 40 s followed by four consecutive frames every 10 s of a tubule emanating from a ring (see *Figure 4—video 1* and *Figure 4—video 2*). Yellow arrows denote appearance of a Cav3 spot on the ring prior to tubulation. White arrows point to the edge of the tubule as it elongates away from the ring. (**C–D**) Gallery of consecutive frames from a time-lapse sequence of a Bin1GFP expressing human myotubes (green). The gallery shows a tubule emanating from a ring (see *Figure 4—video 3* and *Figure 4—video 4*). Time is indicated for each frame as min:s.

The online version of this article includes the following video(s) for figure 4:

**Figure 4—video 1.** Live imaging of Cav3GFP structures on the surface of human myotubes corresponding to *Figure 4A and B*.
https://elifesciences.org/articles/84139/figures#fig4video1

**Figure 4—video 2.** Live imaging of Cav3GFP structures on the surface of human myotubes corresponding to *Figure 4A and B*.
https://elifesciences.org/articles/84139/figures#fig4video2

**Figure 4—video 3.** Live imaging of Bin1GFP structures on the surface of human myotubes corresponding to *Figure 4C and D*.
https://elifesciences.org/articles/84139/figures#fig4video3

**Figure 4—video 4.** Live imaging of Bin1GFP structures on the surface of human myotubes corresponding to *Figure 4C and D*.
https://elifesciences.org/articles/84139/figures#fig4video4

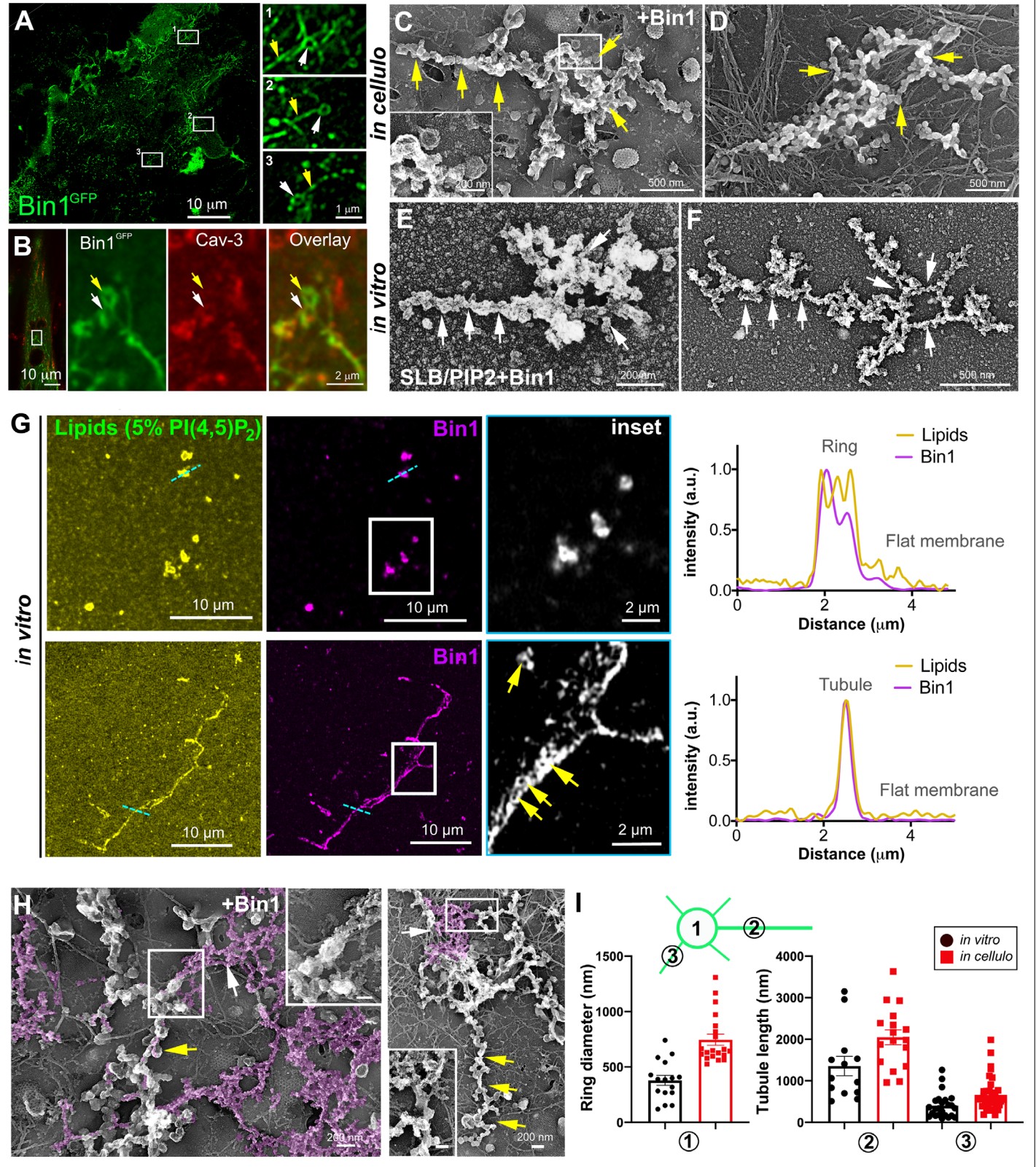

**Figure 5.** Bin1 forms rings and tubes in vitro and in cellulo. (**A**) Super-resolution fluorescence microscopy image of an unroofed human myotube transduced with Bin1GFP (green). Insets numbered 1–3 show the circular organization (white arrowheads) of the structures formed by Bin1 and from which tubes emanate (yellow arrows). (**B**) Super-resolution microscopy image of Bin1+exon11GFP and Cav3 labeling in an intact cultured human myotube. White arrowheads show Cav3 co-localization with Bin1 on both the rings and at the base of the tubules while yellow arrowheads show rings only positive for

*Figure 5 continued on next page*

*Figure 5 continued*

Bin1. (**C–D**) PREM images of unroofed myotubes transduced with Bin1[GFP]. White arrows indicate the central ring-like structure and yellow arrows indicate formation of beaded tubes emanating from the central ring. (**E–F**) PREM images of artificial lipid bilayers incubated with recombinant full-length Bin1. (**G**) Representative airyscan images of supported lipid bilayers (SLBs) containing 5% mol PI4,5P$_2$ and doped with 0.1% of fluorescent lipid dye (DHPE-Oregon green, yellow) and incubated with 1 μM of Bin1-Alexa647 (magenta). Insets show a magnification of Bin1 organization in ring-like structures and tubes (grey) from the corresponding image. Cross-section analysis along the blue dashed line in the related image highlights the intensity profile of Bin1 (magenta) and lipids (yellow) on the flat membrane. Bin1 ring-like organization is present in the membrane and tubular structures (yellow arrows). (**H**) PREM images of unroofed human myotubes transduced with an adenovirus expressing Bin1[GFP]. Yellow arrows show the formation of beaded tubular structures emanating from rings. A Bin1 scaffold similar to the one observed in vitro is pseudo-colored in purple and indicated with white arrows. (**I**) Cartoon of a central ring from which a central pearled tubule and numerous smaller tubules emerge radially. (**1**) Diameter of the rings (in vitro, 16 rings from n=13 images; in cellulo, 20 rings from n=18 images; p<0.0001). (**2**) Length of the longest tubule emanating from Bin1 assemblies (in vitro, 13 tubes from n=13 images; in cellulo, 17 tubes from n=17 images; p<0.05). (**3**) Length of the smaller tubules emanating from Bin1 assemblies (in vitro, 47 small tubes from n=13 images; in cellulo, 72 small tubes from n=18; p<0.05).

The online version of this article includes the following source data and figure supplement(s) for figure 5:

**Source data 1.** Measurements of ring diameters and tube length in vitro.

**Source data 2.** Measurements of ring diameters and tube length in cellulo.

**Figure supplement 1.** Bin1 tubulation assay in mouse myotubes.

**Figure supplement 1—source data 1.** Quantification of tubule density in Bin1-exon11 vs Bin1 +exon 11.

**Figure supplement 1—source data 2.** Quantification of caveolae ring density in Bin1-exon11 vs Bin1 +exon 11.

**Figure supplement 1—source data 3.** Quantification of caveolae ring diameter in Bin1-exon11 vs Bin1 +exon 11.

**Figure supplement 2.** Bin1 forms persistent rings and tubules in vitro.

without prior unroofing (*Figure 7A*). While extremely rare in control myotubes, intact myotubes expressing Bin1 presented a circular organization of 25–40 nm openings characteristic of caveolae necks seen from outside (*Figure 7A, B*). To determine whether the tubules produced at caveolae rings were directly opened to the extracellular environment we performed a penetration assay using the MemBright probe (*Figure 7C*). We simultaneously imaged GFP and MemBright after a brief incubation in myotubes expressing Bin1[GFP]. The membrane invaginations labeled by the probe co-localized with Bin1 labeling demonstrating that the tubules formed by Bin1 are in continuity with the extracellular medium. In agreement with our previous observations, we detected rings that were labelled with both MemBright and Bin1[GFP] (*Figure 7D*).

## Cav3 deficiency alters the formation of tubes induced by Bin1

To test the functional involvement of Cav3 in Bin1-mediated tubulation, we silenced Cav3 expression in human and primary murine myotubes (*Figure 7E, H* and *Figure 5—figure supplement 1*). The 2 siRNAs used enabled a 65–85% decrease in total Cav3 protein leve (*Figure 7F, G*) and resulted in a significant decrease in both the tubulation of membranes formed by Bin1 and the total Bin1 fluorescence (*Figure 7E and H* and *Figure 5—figure supplement 1*). In contrast to controls, Cav3-depleted myotubes formed small Bin1-positive foci at the plasma membrane without extending into tubes confirming the importance of Cav3 structures as initiators of Bin1-induced tubule formation.

We next used three immortalized cell lines derived from patients with caveolinopathies (*Mamchaoui et al., 2011*). The first cell line expresses the P28L mutation in *CAV3*, which usually leads to hyper-CKemia in patients (*Merlini et al., 2002*). The second and third lines express the R26Q mutation which results in a limb-girdle muscular dystrophy 1 C (LGMD-1C) (*Figarella-Branger et al., 2003*) phenotype, Rippling muscle disease (*Betz et al., 2001*) as well as elevated creatine kinase levels (*Carbone et al., 2000*). We produced and analyzed immortalized patient myotubes. The decrease in Cav3 protein levels in differentiated patient myotubes was confirmed using Western blotting (*Figure 8—figure supplement 1*). While we observed a drastic reduction in the amount of Cav3 at the plasma membrane (*Gazzerro et al., 2010*), myotubes were still capable of forming caveolae at the sarcolemma as observed by high-resolution PREM (*Figure 8A–C* and *Figure 8—figure supplement 1*). A Western-blot screen between control and R26Q patient myotubes showed no difference in Bin1 and Junctophilin 2 (SR protein forming contacts between terminal cisternae and T-tubules) expression levels. (*Takeshima et al., 2000*), RyR1 and DHPRα1s subunit (*Figure 8—figure supplement 1*).

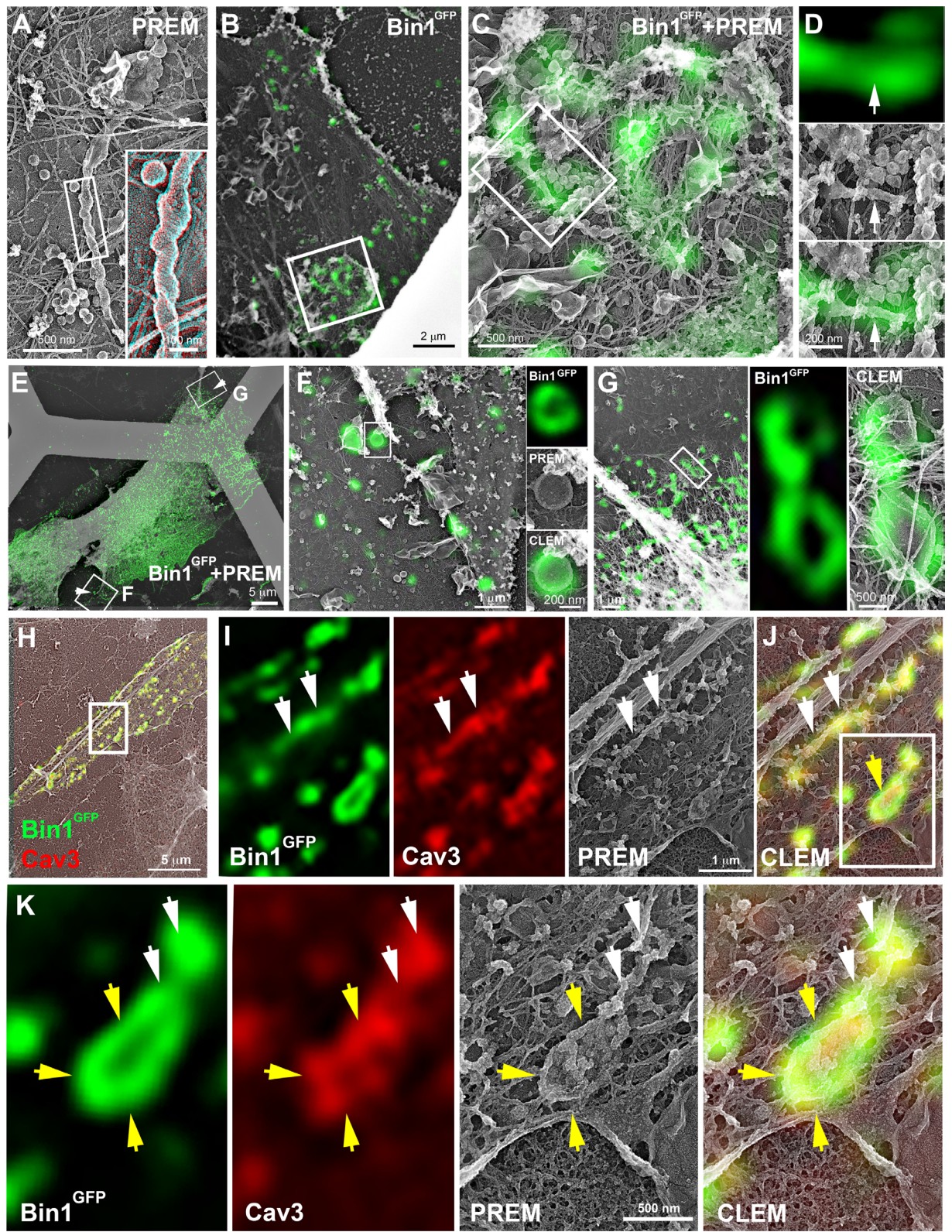

**Figure 6.** CLEM evidence of Bin1 tubules and rings in human and murine myotubes. (**A**) High magnification PREM image of an unroofed human myotube transduced with an adenovirus expressing Bin1GFP forming a characteristic pearled tubule with caveolar material. A 3D anaglyph of the tubule next to a single caveolae with a similar proteinasceous coat is shown in the inset (use red/cyan glasses for viewing). (**B–G**) Correlative LM and PREM images from human myotubes transduced with Bin1GFP (green). (**F and G**) Correlative LM and PREM overlay images corresponding to insets in (**E**). (**H**)

*Figure 6 continued on next page*

*Figure 6 continued*

CLEM overlay of extensively differentiated murine myotubes transduced with Bin1$^{GFP}$ (green) and labeled with antibodies against Cav3 (red). (**I–J**) High magnification PREM views of the inset in (**H**). Bin1$^{GFP}$ and Cav3 labeling on beaded tubes (white arrows) and rings (yellow arrows). (**K**) High magnification views of the inset in (**J**). Bin1$^{GFP}$ (green) colocalized with Cav3 labeling (red) on beaded tubules (white arrows) emanating from rings (yellow arrows).

The online version of this article includes the following video for figure 6:

**Figure 6—video 1.** Correlative super-resolution/PREM for Bin1$^{GFP}$ corresponding to *Figure 6I-K*.

https://elifesciences.org/articles/84139/figures#fig6video1

We tested whether we could find caveolae rings in patient cells. Ultrastructural analysis showed that the spatial organization of caveolae in ring-like structures was altered. We observed instances of deformed rings with supernumerary caveolae (*Figure 8A*). A unique feature of patient myotubes was the production of beaded tubes composed of 5–10 concatenated caveolae not associated with a ring (*Figure 8C* and *Figure 8—figure supplement 1*). These bead-like structures extended from the surface only a couple hundred nanometers inside the muscle fiber suggesting that caveolae have either an increased propensity to fuse with each other or a decreased ability to recruit Bin1 for subsequent tubulation. Quantification of caveolae rings from R26Q patient myotubes showed significantly less caveolae rings at the surface of patient myotubes (in agreement with a decreased Cav3 surface expression) and an increase in their average diameter compared to control myotubes (*Figure 8F and G*). We attempted to rescue this phenotype by re-expressing Cav3. We transduced patient myoblasts with a lentivirus expressing full length Cav3 and differentiated them into myotubes (*Figure 8E–G*). Cav3 re-expression rescued caveolae ring density while reducing the average diameter to control values. We next analyzed Bin1 tubulation in the context of Cav3 deficiency due to the R26Q mutation in patient cells. Myotubes from R26Q patients were transduced with an adenovirus expressing Bin1$^{GFP}$ vector for 24 hr that resulted in a significant decrease in tubule density compared to cells from controls (*Figure 8H–J*). We next analyzed T-tubules in skeletal muscle from two patients from the same family with the Cav3 p.F65S heterozygote mutation (*Figure 8—figure supplement 1*) and one patient with the Cav3 p.D43G heterozygote mutation diagnosed as an LGMD 1 C with Rippling disease and elevated creatine kinase. At the histological level, these patients presented a mild dystrophic phenotype with some necrotic fibers, nuclear internalization, rounded fibers and an almost complete reduction of Cav3 immunostaining at the sarcolemma (*Figure 8—figure supplement 1*). Analysis of skeletal muscle biopsies using thin-section EM tomograms showed characteristic but rare triad abnormalities including T-tubule bifurcations (*Figure 8K–M* and *Figure 8—video 1*) and swelling of the T-tubules (*Figure 8N*).

## Discussion

Pioneer work on early T-tubule formation in muscle has shown that clustered caveolae are present at the T-tubule initiation site (*Franzini-Armstrong, 1991*; *Ishikawa, 1968*; *Parton et al., 1997*; *Schiaffino et al., 1977*) and that both structures present similar lipid composition, that is enriched in cholesterol and sphingolipids (*Parton et al., 1997*). The complexity of the structures involved, their entanglement with the contractile apparatus, but also the lack of tools to visualize nanoscale assemblies have however, limited our understanding of T-tubule biogenesis. Our discovery that caveolae assemble into higher order, 600 nm on average, ring-like structures in mouse and human muscle cells suggests that caveolae may be involved in the formation of membrane platforms on the surface of muscle cells that could participate in T-tubule biogenesis.

To understand the mechanism of T-tubule formation during muscle cell differentiation, we focused on Bin1, both as a marker for T-tubules but also as a membrane sensing and deforming protein through its phosphoinositide-binding (PI) and BAR domains, respectively (*Lee et al., 2002*). In agreement with previous reports, overexpressing Bin1 induced formation of a dense network of membrane tubules (*Nicot et al., 2007*). The use of the MemBright fluorescent probe revealed that these tubes are in direct connection with the external environment. From the outside, we observed numerous 25–40 nm holes formed by the invagination of caveolae, which adopt a ring-like structure. While overexpression of Bin1$^{GFP}$ causes excessive membrane tubulation, Cav3 knock-down lead to a drastic reduction of Bin1-induced tubes, confirming that Cav3-positive caveolae are required for efficient membrane tubulation in mouse and human myotubes. Our results show that Bin1 forms a circular

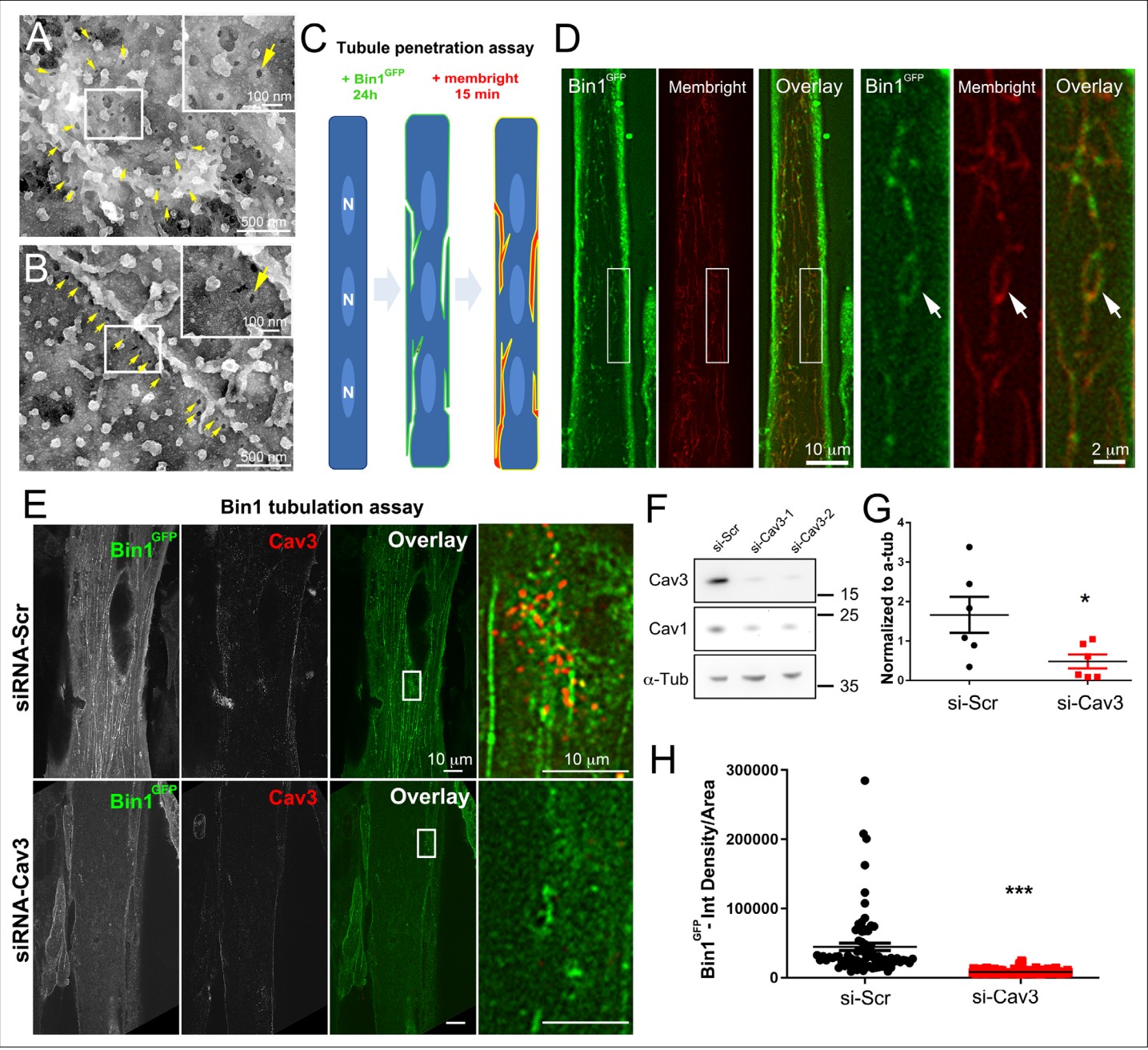

**Figure 7.** Bin1 tubules are in contact with the extracellular medium and depletion of Cav3 decreases Bin1-induced tubulation. (**A–B**) PREM images of intact myotubes transduced with Bin1$^{GFP}$. Yellow arrows indicate necks of caveolae (25–40 nm) seen from the extracellular side of the plasma membrane organized in a circular (**A**) or linear (**B**) fashion. (**C**) Schematic of the MemBright tubule penetration assay. Multi-nucleated myotubes (blue) expressing Bin1$^{GFP}$ are incubated for 15 min with the impermeable lipid probe (red). (N=nucleus). (**D**) Sub-diffracted light microscopy images of cultured murine myotubes transduced with Bin1$^{GFP}$ (green) and incubated with an impermeable lipid probe (red) for 15 min. White arrows in the insets denote ring-like structures positive for both Bin1$^{GFP}$ and MemBright signal. (**E**) Immunofluorescent staining of primary human myotubes at 9 days of differentiation transduced with Bin1$^{GFP}$. In control cells, Bin1$^{GFP}$ overexpression results in abundant membrane tubulation and the appearance of a dense network of tubes. Cav3 deficiency in siRNA-treated myotubes results in a dramatic decrease in tubulation. (**F**) Western-blot analysis of Cav3 and Cav1 protein levels in differentiated myotubes treated with two different siRNAs directed against Cav3 (si-Cav3) and a scramble siRNA (si-Scr). (**G**) Quantification of Cav3 protein levels in cultured myotubes treated with siRNAs directed against Cav3 or scramble siRNA (n=3 independent experiments, p<0.05). (**H**) Quantification of Bin1$^{GFP}$ fluorescence intensity in cultured control and si-Cav3 myotubes transduced with Bin1$^{GFP}$ (si-Scr, 78 myotubes from n=25 images; si-Cav3, 130 myotubes from n=50 images; p<0.0001 from at least three independent experiments).

The online version of this article includes the following source data for figure 7:

*Figure 7 continued on next page*

*Figure 7 continued*

**Source data 1.** Quantification of Cav3 protein levels in myotubes treated with siRNA against Cav3.

**Source data 2.** Quantification of Bin1GFP fluorescence intensity in control and siCav3 myotubes.

**Source data 3.** Western blot uncropped membranes.

scaffold, which subsequently recruits caveolae and become tubulation platforms from which multiple tubules can emanate (*Figure 9*). We propose that there is a physiological mechanism in muscle cells that allows the pre-formation of the structures on which the recruitment of caveolae and their subsequent fusion into a tubule will take place. This suggests that Bin1 could directly bind to surface caveolae in order to tubulate the membrane. In agreement, it was recently shown that cavin 4 directly interacts with Bin1 (*Lo et al., 2021*). We thus postulate that caveolae are recruited on circular Bin1 scaffolds and that these structures grow as additional caveolae are recruited. The platform formed by Bin1, enriched in specific lipids such as phospholipids, cholesterol and sphingolipids, would preferentially recruit Cav3-positive caveolae. Our correlative approach combining PREM to immunofluorescent labeling allowed us to show the presence of Bin1 on caveolae rings as well as Bin1-labeled tubules emanating from rings. CLEM analysis showed that while rings present the characteristic caveolar coat formed by caveolins and cavins at their rim, the omega-shape of caveolae is often lost, suggesting that caveolae are merging with the structure while newly recruited caveolae still accumulate at one of the rings poles. Both light microscopy and EM analyses showed that the tubular elongations can be initiated at several positions on the ring. This is also corroborated by Bin-GFP distribution when overexpressed in myotubes. Time-lapse imaging of the ring dynamics as well as in vitro experiments with recombinant Bin1 demonstrate that multiple tubes can emanate from a single ring. Bin1 BAR domain is responsible for membrane bending and thus for tubulation (*Lee et al., 2002*), as we showed by overexpression in cellulo and in vitro. We propose that tubulation at the start of caveolae rings is induced by Bin1, and one of the accessory proteins present on caveolae might be required to recruit Bin1 to these T-tubule nucleation sites. More recently, it was shown that cavin 4 participates in T-tubule formation by recruiting Bin1 to caveolae. A direct interaction of the proline-rich domain (PRD) of cavin 4 with the SH3 binding domain of Bin1 located at its C-terminus was shown (*Lo et al., 2021*). It was proposed that cavin 4 would have a role in T-tubule membrane remodeling during early development by recycling caveolae from the T-tubule membrane. Interestingly, cavin 4 deletion has been shown to result in formation of pearled tubules. Thus, Bin1 could be recruited at caveolae through a direct protein-protein interaction and newly formed rings could attract additional caveolae.

To understand the contribution of Bin1 to the formation of T-tubules, we turned to in vitro experiments analyzing the self-assembly of recombinant full-length muscle isoform of Bin1 on artificial lipid bilayers enriched in $PI4,5P_2$. In cells, Bin1 formed pearled rings from which tubes emanated, and in in vitro reconstituted lipid bilayers, recombinant Bin1 assembled in very similar ring-like structures that were tightly bound to the substrate. The structures formed by Bin1 resemble the circular structures formed by another BAR domain protein, FCHo2 (*El Alaoui et al., 2022*) that were shown to partition at the edge of clathrin-coated pits and flat lattices (*Sochacki et al., 2017*). We propose that the muscle-specific isoform of Bin1 (full length) has gained the capacity to extend tubules from these rings and this process is facilitated in the presence of Cav3-positive caveolae.

We also found that caveolae rings are entangled with cortical SR cisternae containing RyR1. We initially thought that caveolae rings might have a role in the formation of peripheral coupling sites at the plasma membrane as E-C coupling proteins are predominantly located at the cell periphery during early steps of myotube differentiation. However, electron micrographs in the literature do not show accumulation of caveolae at peripheral coupling sites (*Franzini-Armstrong and Kish, 1995*; *Protasi et al., 1997*). It is thus more likely that contacts between SR cisternae and caveolae rings correspond to the initial sites of T-tubule invagination. The formation of junctions between nascent T-tubules with the SR is on the smooth part of the tubule without any caveolar material. The caveolar coat is still present on the surface of the tubules at the onset of tubulation, but as the immature T-tubules further protrude into the cytoplasm, regions of the tubes appear devoid of caveolae buds. This disappearance coincides with formation of SR and T-tubule junctions when the tubule invaginates deeper into the myotube. We propose that disassembly of caveolar proteins could free the tubule membrane allowing the subsequent formation of junctions with SR proteins such as RyR1. Thus, in agreement

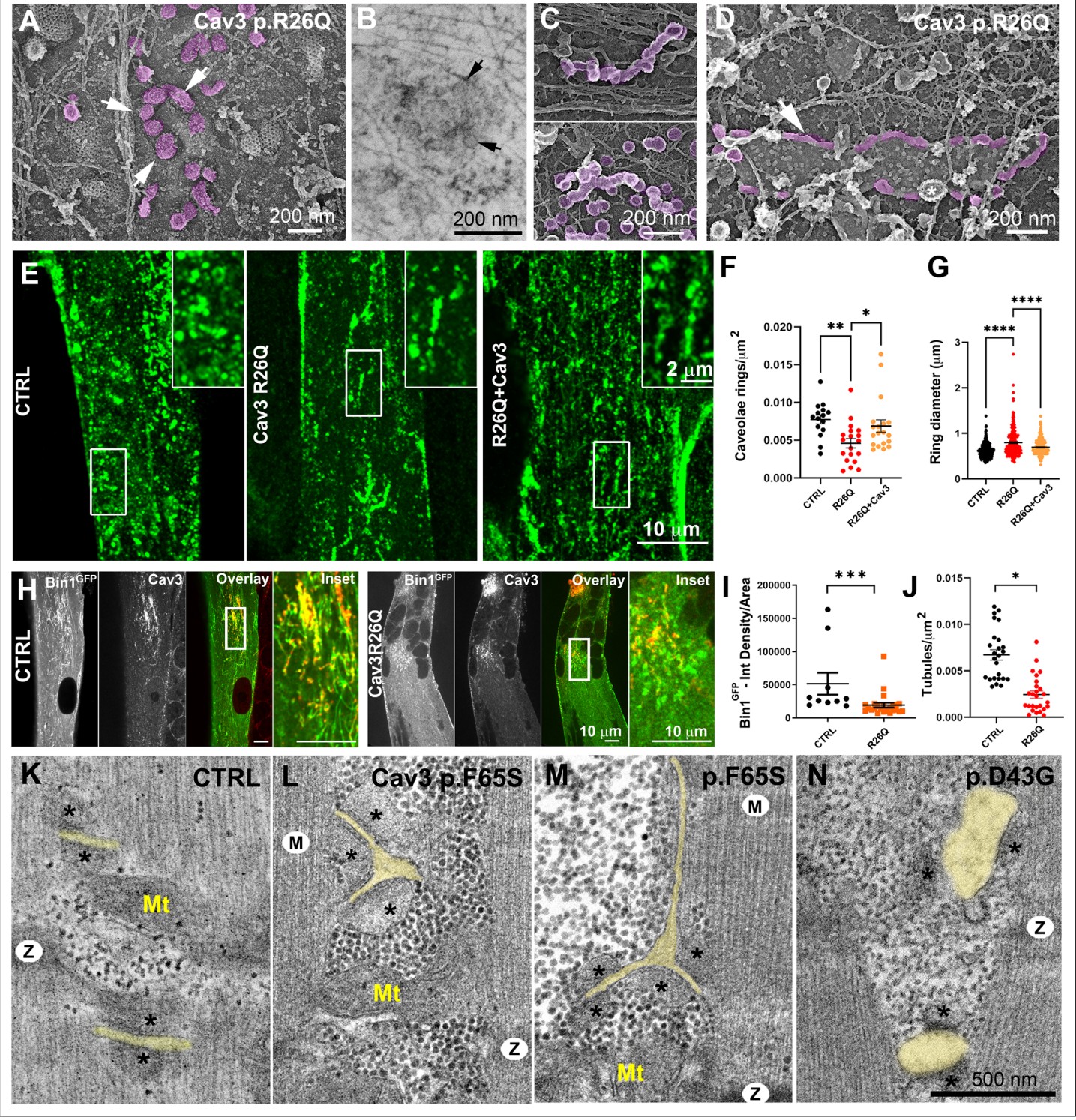

**Figure 8.** Cav3 mutations disorganize caveolae rings in patient-derived cells and T-tubules in patient muscle biopsies. (**A**) High-magnification PREM images of caveolae rings on the cytosolic side of the plasma membrane of sonicated myotubes from patients with the Cav3 R26Q mutation. Caveolae are pseudocolored purple and display an altered structure and spatial organization (white arrows denote individual caveolae with an altered morphology). (**B**) High-magnification view of a caveolae ring from a patient myotube with the R26Q mutation observed by thin-section EM. Black arrows indicate caveolae forming a loose ring-like structure. (**C**) High-magnification PREM images of small tubes made of 5–10 concatenated caveolae in unroofed myotubes from patients with Cav3 R26Q mutation. (**D**) High-magnification view of a giant (>2 μm major axis) caveolae oval-like structure (pseudo-colored in purple) in an unroofed myotube from Cav3 R26Q mutation. White star indicates a clathrin-coated pit. (**E**) Immunofluorescent staining

*Figure 8 continued on next page*

*Figure 8 continued*

of control, R26Q patient myotubes and R26Q patient myotubes expressing full length Cav3. (**F**) Quantification of caveolae ring density from super-resolution images of differentiated control myotubes, caveolinopathy patient myotubes or caveolinopathy patient myotubes with the R26Q mutation stably expressing Cav3[GFP] (CTRL, n=15 myotubes from three independent experiments; R26Q, n=19 myotubes from three independent experiments; R26Q+Cav3, n=19 myotubes from three independent experiments). (**G**) Quantification of caveolae ring diameter from super-resolution images of differentiated control myotubes, caveolinopathy patient myotubes or caveolinopathy patient myotubes with the R26Q mutation stably expressing Cav3[GFP] (CTRL, n=310 rings from three independent experiments; R26Q, n=219 rings from three independent experiments; R26Q+Cav3, n=670 rings from three independent experiments). (**H**) Immunofluorescent staining of control and Cav3 R26Q patient myotubes transduced with Bin1[GFP]. In control cells, Bin1[GFP] overexpression results in excessive membrane tubulation and appearance of a dense network of tubes. Cav3 deficiency in patient myotubes results in a dramatic decrease in tubulation. (**I**) Quantification of Bin1[GFP] fluorescence intensity in cultured control and patient myotubes transduced with Bin1[GFP]. The Cav3 R26Q mutation results in a significant decrease in Bin1 labeling. (CTRL, 10 myotubes from n=8 images; Cav3 R26Q, 24 myotubes from n=16 images; p<0.001 from at least three independent experiments). (**J**) Quantification of tubule density from super-resolution images of differentiated myotubes either expressing Bin1-exon11[GFP] or Bin1+exon11[GFP] (Bin1-exon11[GFP], n=21 myotubes from three independent experiments; Bin1+exon11[GFP], n=14 myotubes from three independent experiments; p<0.001). (**K–N**) High magnification thin-section EM images of muscle biopsies from a control subject (**K**) and patients with either Cav3 F65S (**L and M**) or D43G mutations (**N**). SR terminal cisternae are denoted with black asterisks. T-tubules are pseudo-colored pale yellow. Mitochondria (Mt), Z-Disk (**Z**), M-band (**M**). In (**L**) and (**M**) T-tubules form abnormal bifurcations. Instances of T-tubule swelling (**N**) are also observed in patient biopsies.

The online version of this article includes the following video, source data, and figure supplement(s) for figure 8:

**Source data 1.** Quantification of caveolae ring density ctrl vs R26Q vs R26Q+Cav3-GFP.

**Source data 2.** Quantification of caveolae ring diameter in ctrl vs R26Q vs R26Q+Cav3-GFP.

**Source data 3.** Quantification of tubule density in control and patient myotubes transduced with Bin1 +ex11.

**Source data 4.** Quantification of tubule density in control and patient myotubes transduced with Bin1exon11.

**Figure supplement 1.** Cav3 protein levels and characteristic ultrastructure of beaded caveolae tubes in caveolinopathy patient myotubes and characteristic histology of caveolinopathy patient muscle biopsies.

**Figure supplement 1—source data 1.** Densitometric quantification of Cav3 protein levels in patient myotubes.

**Figure supplement 1—source data 2.** Western blot uncropped membranes.

**Figure supplement 1—source data 3.** Western blot uncropped membranes.

**Figure 8—video 1.** Tomogram from a semi-thin section of a caveolinopathy patient presenting a bifurcation and corresponding to *Figure 8M*.
https://elifesciences.org/articles/84139/figures#fig8video1

with observations from zebrafish (*Lo et al., 2021*), our results support the notion that part of the caveolar material must be removed from the tubule and/or diffuse along the tubule membrane for functional junctions to form.

Lastly, our analysis of the link between caveolae rings and tubulation by Bin1 in cells from patients with *CAV3* mutations supports a role for Cav3-positive caveolae in the formation of a functional E-C coupling system. While caveolinopathy patients form overall normal triads, their muscles present aggregation of Cav3 at the Golgi apparatus leading to a drastic decrease in the amount of Cav3 at the plasma membrane as well as defects in T-tubule morphology which may explain in part the occurrence of myalgias and exercise intolerance characteristic of these patients (*Gazzerro et al., 2010*; *Minetti et al., 2002*; *Brauers et al., 2010*; *Sotgia et al., 2003*). However, in agreement with previous reports (*Dewulf et al., 2019*), our PREM analysis of patient myotubes showed that some caveolae are still present at the plasma membrane albeit with an altered ultrastructure suggesting that some Cav3 or Cav1 oligomers do reach the sarcolemma despite their intracellular retention. Importantly, caveolae ring formation as well as tubulation by Bin1 were defective in patient myotubes. Presumably, the mixed Cav3 population (wild type and mutant) in caveolae could prevent enrichment of DHPR and reduce Bin1 tubulation. In agreement, it was shown that myotubes from caveolinopathy patients present a reduction in both depolarization-induced Ca$^{2+}$ release and influx (*Ullrich et al., 2011*) with disarrays in the colocalization of the DHPR and RyR1, thereby reducing the efficiency of excitation-contraction coupling. Alternatively, mutated Cav3 may alter caveolae fusion with Bin1 rings and thus alter tube initiation or might not recycle efficiently from the nascent T-tubule and thereby inhibit the formation of SR-T-tubule junctions. Importantly, these mechanisms could also contribute to biogenesis of T-tubules in the heart as Cav3, cavin 1 and cavin 4 mutations lead to cardiomyopathies (*Rajab et al., 2010*; *Rodriguez et al., 2011*; *Vatta et al., 2006*).

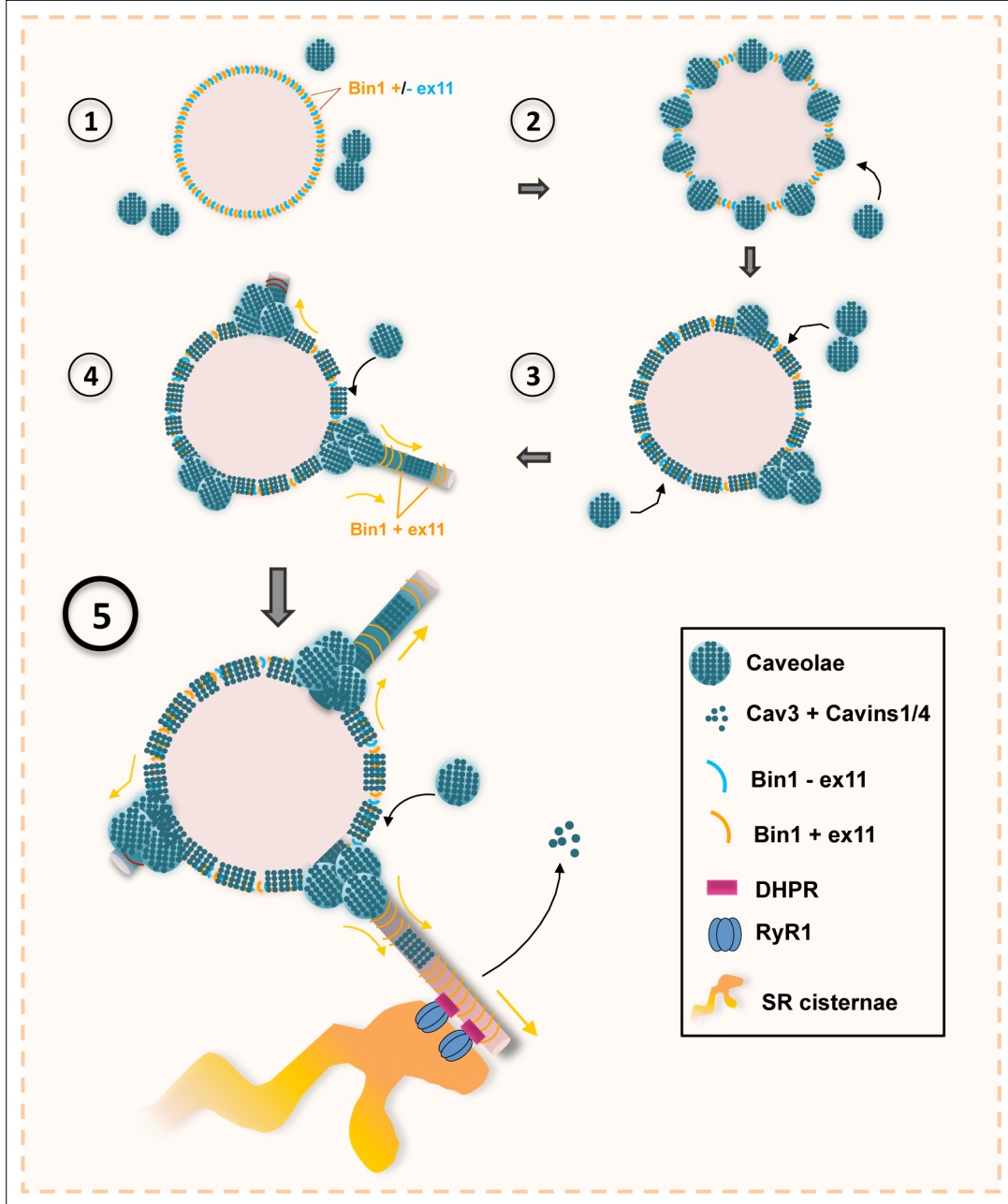

**Figure 9.** Model of Bin1 and Cav3-mediated ring formation and tubulation. (1) Bin1 molecules polymerize into ring-like structures and recruit Cav3-positive caveolae. (2) Caveolae assemble into circular structures while additional caveolae accumulate at specific spots on the ring periphery. (3) Bin1-positive tubules emanate from regions where caveolae accumulate. (4-5) Tubules containing the α1s-subunit of the DHPR elongate from the central ring and start forming contacts with RyR1-positive SR cisternae. Caveolar material is either removed from the tubule or diffuses along the tubule membrane.

Overall, we described caveolae rings, induced by recruitment of caveolae on circular Bin1 scaffolds at the plasma membrane, as essential structures for the initiation of T-tubule formation. We show that Cav3 deficiency leads to defects in caveolae ring formation as well as defects in Bin1 tubulation. Our results support a model where disassembly of caveolar proteins is required for establishment of triads and formation of the molecular complex responsible for excitation-contraction coupling. Thus, our discovery of caveolae rings could be the missing link for the initial steps of T-tubule formation and

**Table 1.** List of primary antibodies.

| Antibodies | Provider | Product ref |
|---|---|---|
| Cav3 (Mouse) monoclonal | BD Biosciences | 610421 |
| Cav3 (Rabbit) polyclonal | Abcam | ab2912 |
| Cav-1 (Rabbit) polyclonal | Santa Cruz | sc894 |
| Cavin 4 (MURC) (Rabbit) polyclonal | Merck | HPA020973 |
| DHPR (Mouse) monoclonal | Abcam | ab2862 |
| RyR1 (Rabbit) polyclonal | Custom | *Marty et al., 1994* |
| Bin1 (Mouse) monoclonal | Merck | 05-449-C |
| Bin1 (Rabbit) polyclonal | Custom | *Nicot et al., 2007* |
| GFP (Rabbit) polyclonal | Thermofischer | A11122 |
| -actinin 2(Mouse) monoclonal | Sigma Aldrich | A7811 |
| Junctophilin 2 (Rabbit) polyclonal | Thermofischer | PA5-20642 |
| GAPDH (Rabbit) polyclonal | Santa Cruz | sc25778 |
| -tubulin (Mouse) monoclonal | Thermofischer | 236-10501 |

provides the basis for a precise characterization of T-tubule biogenesis in healthy skeletal muscle and in the pathophysiology of caveolinopathies.

## Materials and methods
### Antibodies
Primary antibodies are listed in *Table 1*. Secondary antibodies for immunofluorescence were Alexa Fluor 488, Alexa Fluor 568, and Alexa Fluor 647 conjugates (Life Technologies, France). Secondary antibodies for Western blotting were coupled to horseradish peroxidase (HRP; Jackson Laboratories, USA).

### Human and murine myoblast cultures
Primary mouse skeletal muscle cells were prepared from 3- to 4-day-old mouse pups. Cells were maintained in tissue culture dishes coated with Matrigel matrix at 1:100 (Corning, France) in DMEM medium with 20% fetal bovine serum (FBS), 50 U/ml penicillin, 50 mg/ml streptomycin (growth medium), and 1% chicken embryo extract (Seralab, UK). Differentiation was induced when cells were ~80% confluent by switching to differentiation medium; DMEM medium with 2% horse serum (Life Technologies, France) and 80 ng/mL of agrin (R&D Systems, USA). To avoid detachment due to strong contractions and to keep cells in culture for prolonged periods of differentiation (up to 10 days), myotubes were covered with a layer of Matrigel Growth Factor Reduced Basement Membrane Matrix, Phenol Red-Free at 1:3 (Corning, France) (*Falcone et al., 2014*).

Human muscle biopsies (control muscle) used in this study were obtained via Myobank-AFM, affiliated with EuroBioBank, in accordance with European recommendations and French legislation (authorization AC-2019–3502). Human immortalized control and patient myoblast cell lines (Cav3 P28L, Cav3 R26Q) were grown in proliferation medium: 1 volume of M199 (Life Technologies, France), 4 volumes of DMEM Glutamax, 20% fetal bovine serum, 50 µg/mL gentamicin (Life Technologies, France), 25 µg/mL fetuin (Life Technologies, France), 0.5 ng/mL basic fibroblast growth factor (Life Technologies, France), 5 ng/mL human epidermal growth factor (Life Technologies, France), 0.2 µg/mL dexamethasone (Sigma, France), 5 µg/mL insulin (Sigma, France).

**Table 2.** List of siRNA sequences.

| Target | Human siRNASequence |
|---|---|
| Cav3 (1) | 5'-CAGAUCUCGAGGCCCAGAUCG-3' |
| Cav3 (2) | 5'-AAGCACAAUGGCCCUUCGCUC-3' |
| Target | Murine siRNASequence |
| Cav3 | 5'-GGUUCCUCUCAAUUCCAC-3' |

Differentiation is induced 48 hr after seeding using a differentiation medium consisting of DMEM Glutamax, 5% horse serum, 50 µg/mL gentamicin, supplemented with 5 µg/mL insulin (Sigma, France). Agrin (R&D Systems, USA) was added at 80 ng/mL after 2 days of differentiation.

Cells were transduced with lentiviral vectors expressing Cav3 and a GFP reporter gene. Myoblasts were transduced with an MOI of 5, and cells were differentiated into myotubes.

### siRNA-mediated knock-down

For siRNA treatment, myotubes (differentiated for either 2 or 4 days) were transfected twice for 48 hr using 200 nM siRNA and Lipofectamine RNAiMax transfection reagent (Life Technologies, France) according to the manufacturer's instructions. Targeting and control siRNAs were synthesized by Euro-gentec, Belgium. A list of siRNAs used and sequences can be found in (*Table 2*).

### Adenovirus Bin1$^{GFP}$ and Bin1-Δexon11$^{GFP}$ transduction

Differentiating myotubes were transduced with adenoviral vectors expressing Bin1 protein including exon 11 (Bin1$^{GFP}$) and Bin1 isoform without exon 11 (Bin1-Δexon11$^{GFP}$) with a MOI of 75. The viruses are plated and incubated on the cells for 6 hr at 37 °C in DMEM Glutamax medium prior to changing to differentiation medium.

### Protein purification and immunofluorescent labeling

Human isoform of amphiphysin 2/BIN1 including exon 11 was expressed in Rosetta 2 bacteria and purified by affinity chromatography using glutathione Sepharose 4B beads according to the manufacturer's instructions (GE Healthcare) in 50 mM Tris at pH 8.0, 100 mM NaCl. Proteins were expressed overnight at 18 °C using 1 mM IPTG induction, dialyzed overnight in a Slide-A-Lyzer dialysis cassette (MWCO 10,000) before Alexa Fluor 488 or 647 maleimide labeling (Invitrogen). Protein concentrations were measured using Bradford assay kits (Biorad).

### Lipids and reagents

Natural phospholipids including Egg-phosphatidylcholine (PC), Brain-phosphatidylserine (PS), Brain-PI4,5P$_2$, were from Avanti Polar Lipids, Inc Oregon Green 488 DHPE and Alexa Fluor 647 Maleimide labelling kit from Invitrogen.

### Supported lipid bilayers

Lipid mixtures consisted of 85% Egg-PC, 10% Brain-PS and 5% of Brain-PI4,5P$_2$. Fluorescent lipids were added to 0.2%. Supported lipid bilayers for fluorescence microscopy and PREM experiments were prepared from large unilamellar vesicles (LUVs, diameter ~100 nm) as described (*Braunger et al., 2013*). Experiments were performed by injecting 20 µL of buffer (20 mM Tris, pH 7.4, 150 mM NaCl) containing 1 µM of non-labeled Bin1 (for PREM experiments) or 1–2 µM of Bin1-Alexa 647 (for LM experiments). Additionally, 0.5 mg/ml of casein was added to the buffer for fluorescence microscopy imaging. Supported lipid bilayers were imaged on a Zeiss LSM880 Airyscan confocal microscope (MRI facility, Montpellier). Excitation sources used were an Argon laser for 488 nm and a Helium/Neon laser for 633 nm. Acquisitions were performed on a 63 x/1.4 objective. Multidimensional acquisitions were acquired via an Airyscan detector (32-channel GaAsP photomultiplier tube array detector).

### Immunoblot analysis

Cell samples were collected using an NaCl (150 mM)-EDTA (10 mM) buffer with added proteinase inhibitor cocktail 1:100 (Sigma-Aldrich, France) and then denatured in Laemmli denaturing buffer 4 X. Protein samples were separated by electrophoresis (4–12% bis-acrylamide gel; Life Technologies, France), electrotransfered to 0.45 µm nitrocellulose membranes (Life Technologies, France) and labeled with primary antibodies then secondary antibodies coupled to HRP. The presence of proteins in samples was detected using Immobilon Western Chemiluminescent HRP Substrate (Sigma-Aldrich, France). Acquisition was performed on a ChemiDoc Imaging System (Biorad, Inc, France).

### Immunofluorescence microscopy

Mouse and human cells were grown on glass coverslips, washed in warm PBS, fixed in paraformaldehyde (4% in PBS, 15 min), and then washed in PBS, permeabilized (10 min, 0.5% Triton X-100 in

PBS), and blocked (5% BSA in PBS with 0.1% Triton X-100, 30 min). Antibody labeling was performed by addition of 200 μL blocking solution with primary or secondary antibodies and washing with PBS with 0.1% Triton X-100. Samples were mounted in Vectashield containing DAPI (Vector Laboratories, USA). Images were acquired using a Nikon Ti2 microscope, driven by Metamorph (Molecular Devices), equipped with a motorized stage and a Yokogawa CSU-W1 spinning disk head coupled with a Prime 95 sCMOS camera (Photometrics) equipped with a 100 x oil-immersion objective lense. Super-resolution images were obtained using the LiveSR module (Gataca Systems). DAPI, Alexa Fluor 488, Alexa Fluor 568 and Alexa Fluor 647 were sequentially excited. Z-series from the top to the bottom of fibers were sequentially collected for each channel with a step of 0.1–0.3 μm between each frame. Image quantification was performed using National Institutes of Health's FIJI (*Schindelin et al., 2012*).

### MemBright live imaging

Myotubes differentiated between 7 and 9 days were placed in an incubation chamber to be maintained at 37 °C and 5% $CO_2$ throughout the experiment. At the time of observation, the medium was changed to DMEM Glutamax and 0.1 μM of MemBright 640 (Cytoskeleton, Inc, France). Similar to immunofluorescent staining the cells were observed using the Nikon Ti2 spinning disk confocal microscope through a x100 immersion objective and z-stacks (300 nm) were acquired with a thickness of 5 μm for 20 min with a stack every 3 min. The generated images were then analyzed using FIJI.

### Time-lapse imaging

Myotubes differentiated between 7 and 9 days were placed in an incubation chamber and maintained at 37 °C and 5% $CO_2$ throughout the experiment. Cells were observed using the Nikon Ti2 spinning disk confocal microscope (equipped with a SR module) through a x100 immersion objective and frames were acquired every 10 s during periods ranging from 15 min to 3 hr. The generated images were then analyzed using FIJI and movies played back at 5 frames per second (*Figure 4—videos 1–4*).

### Platinum-replica EM of unroofed myotubes

Adherent plasma membranes from cultured cells grown on glass coverslips were obtained by sonication as described previously (*Heuser, 2000*). Sample processing for platinum-replica EM of unroofed cells was performed as follows: 2% glutaraldehyde/2% paraformaldehyde-fixed cells were further sequentially treated with 0.5% OsO4, 1% tannic acid, and 1% uranyl acetate before graded ethanol dehydration and hexamethyldisilazane (HMDS) substitution (LFG Distribution, France). Dried samples were then rotary shadowed with 2 nm of platinum (sputtering) and 4–6 nm of carbon (carbon thread evaporation) using an ACE600 metal coater (Leica Microsystems, Germany). The resultant platinum replica was floated off the glass with hydrofluoric acid (5%), washed several times on distilled water, and picked up on 200 mesh formvar/carbon-coated EM grids. The grids were mounted in a eucentric side-entry goniometer stage of a transmission electron microscope operated at 120 kV (JEOL, Japan), and images were recorded with a Xarosa digital camera (EM-SIS, Germany). Images were processed in Adobe Photoshop to adjust brightness and contrast and presented in inverted contrast.

### Histomorphological and ultrastructural analyses

Human muscle biopsies from two patients carrying the Cav3 mutation p.F65S, one patient carrying the Cav3 mutation p.D43G, and one healthy control muscle were performed at the Centre de Référence de Pathologie Neuromusculaire Paris-Est (Institut de Myologie, GHU Pitié-Salpêtrière, Paris, France), following written informed consent specially dedicated for diagnosis and research. Muscle was frozen in liquid nitrogen-cooled isopentane. For all imaging, exposure settings were identical between compared samples and viewed at room temperature. For conventional histochemical techniques on human biopsies, 10-μm-thick cryostat sections were stained with antibodies against Cav3, Hematoxylin and eosin or with reduced nicotinamide adenine dinucleotide dehydrogenase-tetrazolium reductase by standard methods. Pictures of each section were obtained with a Zeiss AxioCam HRc linked to a Zeiss Axioplan Bright Field Microscope and processed with the Axio Vision 4.4 software (Zeiss, Germany).

For ultrastructural analysis of patient biopsies, fresh muscle samples were fixed in glutaraldehyde (2.5%, pH 7.4), postfixed in osmium tetroxide (OsO4, 2%), and embedded in resin (EMBed-812,

Electron Microscopy Sciences, Hatfield, PA). Ultra-thin (80 nm) sections were stained with uranyl acetate and lead citrate.

For ultrastructural analysis of cultured myotubes, cells were fixed with 2% paraformaldehyde, 2% glutaraldehyde in 0.1 M phosphate buffer (pH 7.4). Samples were post-fixed with 2% OsO4, in 0.1 M phosphate buffer (pH 7.4) for 1 hr, dehydrated in a graded series of acetone including a 1% uranyl acetate staining step in 70% acetone, and finally embedded in epoxy resin (EMBed-812, Electron Microscopy Sciences, USA). Ultra-thin (70 nm) sections were stained with uranyl acetate and lead citrate. Observations were made on a transmission electron microscope operated at 120 kV (JEOL, Japan), and images were recorded with a Xarosa digital camera (EM-SIS, Germany).

## Unroofing and PREM immunocytochemistry

Unroofing was performed by sonication. Coverslips were quickly rinsed three times in Ringer + Ca (155 mm NaCl, 3 mm KCl, 3 mm NaH2PO4, 5 mm HEPES, 10 mm glucose, 2 mm CaCl2, 1 mm MgCl2, pH 7.2), then immersed 10 s in Ringer-Ca (155 mm NaCl, 3 mm KCl, 3 mm NaH2PO4, 5 mm HEPES, 10 mm glucose, 3 mm EGTA, 5 mm MgCl2, pH 7.2) containing 0.5 mg/mL poly-l-lysine, then quickly rinsed in Ringer-Ca then unroofed by scanning the coverslip with rapid (2–5 s) sonicator pulses at the lowest deliverable power in KHMgE buffer (70 mm KCl, 30 mm HEPES, 5 mm MgCl2, 3 mm EGTA, pH 7.2).

Unroofed cells were immediately fixed in KHMgE: 4% PFA for 10 min for light microscopy, 4% PFA for 45 min for PREM of immunogold-labeled samples, 2% PFA–2% glutaraldehyde for 10–20 min for PREM.

## Correlative-EM of unroofed myotubes

For correlative-EM on unroofed cells, myotubes were grown on alpha-numerically gridded bottom dishes (Ibidi, France). Adherent plasma membranes were obtained by sonication and were immediately immersed in 4% paraformaldehyde and the proteins of interest were then labeled by immunofluorescence in saturation buffer (1% BSA in KHMgE buffer). Immunofluorescent stainings were acquired with a Nikon Ti2 microscope, and super-resolution images were obtained with the LiveSR module (Gataca Systems). Observations were performed at different magnifications in order to optimize the localization of the myotubes of interest (4x, 20x and 100x objective lenses). The samples were then incubated in 2% glutaraldehyde at 4°C overnight and treated sequentially with 0.5% Osmium tetroxide OsO4, 1.5% tannic acid and 1% uranyl acetate before being dehydrated with ethanol and HMDS (LFG Distribution, France). Dried samples were then rotary shadowed with 2 nm of platinum and 4-6 nm of carbon using the ACE600 (Leica Microsystems, Germany). Areas of interest from the resulting platinum replicas were separated from glass with 5% hydrofluoric acid and deposited on carbon-coated EM grids (200 mesh formvar/carbon, LFG Distribution, France). Images were processed in Adobe Photoshop to adjust brightness and contrast, and unroofed PMs are presented in inverted contrast. Super-resolution microscopy and electron microscopy images were overlaid using Adobe Photoshop software.

## Anaglyphs and Tomograms

Anaglyphs were made by converting the –10° tilt image to red and the +10° tilt image to cyan (blue/green), layering them on top of each other using the screen blending mode in Adobe Photoshop, and aligning them to each other. Tomograms were made by collecting images at the tilt angles up to ±25° relative to the plane of the sample with 5° increments. Images were aligned by layering them on top of each other in Adobe Photoshop. For PREM tomograms, the sample in the TEM holder was tilted with an angle of ±20° and an image was taken every 5°. The brightness and contrast of the images were adjusted in Adobe Photoshop and the images were presented in reverse contrast. Movies were made by combining LM, PREM images produced by increasing magnifications of the same myotube, and tomograms of the highest magnification region using Adobe Photoshop and FIJI. For muscle biopsy tomograms, semi-thin sections (200 nm) were contrasted with uranyl acetate (5 min) and lead citrate (5 min); the images were acquired with Radius software and the TEM holder was tilted with an angle of ±60° and an image was taken every 2°; the alignment and reconstruction of the tilted series were carried out with the TomoJ plugin (*Messaoudii et al., 2007*; *Sorzano et al., 2020*) of ImageJ software.

## Data analysis and statistics

Graphs and statistical analyses were performed with Excel and GraphPad Prism v. 6.00 software. Values are expressed as means ± SEM. The number of samples (n), representing the number of independent biological replicates, is indicated in the figure legends. Statistical comparisons between two groups were performed using unpaired one- or two-tailed Student's t tests as specified. Statistical tests applied are indicated in the figure legends. $p < 0.05$ was considered statistically significant. In all statistical analyses, the levels of significance were defined as: $*p < 0.05$, $**p < 0.01$, $***p < 0.001$ and $****p < 0.0001$.

## Study approval

Animal studies conform to the French laws and regulations concerning the use of animals for research and were approved by an external ethics committee (approval no. 00351.02 delivered by the French Ministry of Higher Education and Scientific Research). For human studies, all individuals provided informed consent for muscle biopsies according to a protocol approved by the ethics committee of the Centre de Référence de Pathologie Neuromusculaire Paris-Est, Institut de Myologie, Assistance Publique-Hôpitaux de Paris, GH Pitié-Salpêtrière, Paris, France.

## Acknowledgements

We thank Nicolas Charlet-Berguerand, Christophe Lamaze and Jocelyn Laporte for reagents, Andrée Rouche and Teresinha Evangelista for helpful discussions. We also thank the IBPS electron microscopy platform (Sorbonne University, Paris, France), the MyoBank-AFM and the Myoline platform from the Institute of Myology (Paris, France). This work has been funded by Sorbonne Université, INSERM, Association Institut de Myologie core funding, the Agence Nationale de la Recherche (grants ANR-21-CE13-0018-01 to SV, ANR-18-CE17-0006-02 to MB) and Association Française contre les Myopathies (AFM-Telethon).

## Additional information

### Funding

| Funder | Grant reference number | Author |
| --- | --- | --- |
| Agence Nationale de la Recherche | ANR-21-CE13-0018-01 | Stéphane Vassilopoulos |
| Agence Nationale de la Recherche | ANR-18-CE17-0006-02 | Marc Bitoun |
| Association Française contre les Myopathies | | Marc Bitoun |

The funders had no role in study design, data collection and interpretation, or the decision to submit the work for publication.

### Author contributions

Eline Lemerle, Conceptualization, Formal analysis, Investigation, Visualization, Methodology, Writing - original draft; Jeanne Lainé, Bruno Cadot, Laura Picas, Formal analysis, Investigation, Methodology, Writing - review and editing; Marion Benoist, Formal analysis, Methodology; Gilles Moulay, Anne Bigot, Clémence Labasse, Angéline Madelaine, Alexis Canette, Perrine Aubin, Norma B Romero, Isabelle Marty, Formal analysis, Investigation, Methodology; Jean-Michel Vallat, Vincent Mouly, Resources, Formal analysis, Investigation, Methodology; Marc Bitoun, Formal analysis, Funding acquisition, Investigation, Methodology, Writing - review and editing; Stéphane Vassilopoulos, Conceptualization, Formal analysis, Supervision, Funding acquisition, Validation, Investigation, Visualization, Methodology, Writing - original draft, Project administration

### Author ORCIDs

Alexis Canette http://orcid.org/0000-0002-2750-6568
Laura Picas http://orcid.org/0000-0002-5619-5228

Stéphane Vassilopoulos ⬤ http://orcid.org/0000-0003-0172-330X

### Ethics

For human studies, all individuals provided informed consent for muscle biopsies according to a protocol approved by the ethics committee of the Centre de Référence de Pathologie Neuromusculaire Paris-Est, Institut de Myologie, Assistance Publique-Hôpitaux de Paris, GH Pitié-Salpêtrière, Paris, France.

Animal studies conform to the French laws and regulations concerning the use of animals for research and were approved by an external ethics committee (approval no. 00351.02 delivered by the French Ministry of Higher Education and Scientific Research).

### Decision letter and Author response

Decision letter https://doi.org/10.7554/eLife.84139.sa1
Author response https://doi.org/10.7554/eLife.84139.sa2

## Additional files

### Supplementary files

• MDAR checklist

### Data availability

All datasets supporting the findings of this study are available in Dryad at https://doi.org/10.5061/dryad.k98sf7m98.

The following dataset was generated:

| Author(s) | Year | Dataset title | Dataset URL | Database and Identifier |
| --- | --- | --- | --- | --- |
| Vassilopoulos S, Lemerle E, Lainé J, Benoist M, Moulay G, Bigot A, Labasse C, Madelaine A, Canette A, Aubin-Tessier P, Vallat J, Romero NB, Bitoun M, Marty I, Mouly V, Cadot B, Picas L | 2023 | Data from: Caveolae and Bin1 form ring-shaped platforms for T-tubule initiation | https://dx.doi.org/10.5061/dryad.k98sf7m98 | Dryad Digital Repository, 10.5061/dryad.k98sf7m98 |

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
