## [Editor Report]

Lemerle et al. provide convincing evidence that advances our fundamental understanding of how transverse tubules may be formed, a significant gap in our understanding of excitation contraction coupling and muscle biology more broadly. They utilize advanced correlative light and electron microscopy and molecular biology approaches to demonstrate the presence of Bin1 and caveolae containing rings that are capable and necessary to properly tubulate membranes in developing striated muscle.

---

## [Decision Letter]

**Decision letter after peer review:**

Thank you for submitting your article "Caveolae and Bin1 form ring-shaped platforms for T-tubule initiation" for consideration by *eLife*. Your article has been reviewed by 3 peer reviewers, including Benjamin L Prosser as the Reviewing Editor and Reviewer #1, and the evaluation has been overseen by Suzanne Pfeffer as the Senior Editor. The following individual involved in the review of your submission has agreed to reveal their identity: Michael Ibrahim (Reviewer #3).

Essential revisions:

The reviewers have a generally enthusiastic consensus about the potential impact of the work and its high significance to the field of striated muscle biology, barring appropriate revisions.

We recommend that the authors focus on the following essential revisions:

1) More rigorous quantification of imaging-based observations, with specific concerns outlined below.

2) Strengthening the section on pathological implications of Cav3 variants, which currently is insufficiently supported. This could include more rigorous validation of disease phenotypes and/or rescue experiments with the restoration of Cav3 levels and tubule morphology.

*Reviewer #1 (Recommendations for the authors):*

The major issue that reduces the strength of several conclusions throughout the manuscript is the lack of quantification or appropriate controls for several imaging-based observations. Additional quantification throughout would add important context regarding the frequency, consistency, and effect size of key observations, and how they change with different molecular manipulations (such as Bin1 overexpression) or pathological variants. While the authors may have a very trained eye for detecting structural differences via EM, they cannot assume a broad readership will, and representative images are insufficient to support robust conclusions. The following 5 points are all related to this theme:

1. The conclusion that Bin1GFP induced the formation of rings and tubules while the removal of exon 11 prevented tubule formation is insufficiently supported. Some quantification of ring and tubule prevalence should be provided.

2. For figure 7A-B, necks of caveolae are seen on the extracellular surface of intact cells overexpressing Bin1, but no similar EM of a non-Bin1 overexpressing cell or quantification is presented to show such structures do not exist or exist at lower frequency in the absence of Bin1 (or with Bin1exon11 mutant).

3. For Figure 8A-C, it is claimed that the EM images show altered structure and spatial organization of caveolae rings with rare supernumerary caveolae in R26Q variants. This is not clear nor particularly interpretable without some sort of quantification of these morphological changes and their occurrence. It is not clear whether the reduction in Cav3 protein results in altered rings, or simply fewer rings, and this is an important distinction for understanding pathomechanisms.

4. For Figure 8H-K, there must also be some quantification of the observed triad abnormalities, and again a comparison to control samples, to have this be a meaningful contribution. Taken together with the prior point, this limits any insight into pathology, and the claim in discussion (line 436) that caveolae ring formation is defective in patient myotubes is insufficiently supported.

5. The addition of actin filaments in the model – particularly in the early stages of ring formation- seems speculative and potentially misleading. The authors' in vitro data in lipid bilayers (that presumably lack actin filaments) show that actin is dispensable for ring formation and even early tube formation, and actin is not mentioned in the main text. I would advise removing it from the cartoon.

*Reviewer #2 (Recommendations for the authors):*

This manuscript is very interesting and the data are compelling. I do however have some additional comments below for the authors' consideration.

The authors should clarify the statistics used. They state that 'n' is the number of independent biological replicates, but it is not always clear what the biological replicate is. For example, in Figure 1B is 'n' the ring or the myotube, in Fig1J is it the caveolae or the experimental repeat (which I think it should be)? It would be useful for this clarity throughout.

In Figure 5G do the conditions or the Bin1 isoform vary between the upper and lower panels? Can this be added to the legend? It is not quite clear to me what is being shown in Figure 6E.

The legend for Figure 6B-E is identical to 6F – could they be combined to simplify? Similarly for Figures6G and H?

Lines 288-289 suggest BIN1 is not always localised to caveolae but the authors might consider if it would be more convincing for the data cited to include cav3 labelling.

In Fig7A are the authors suggesting the openings are caveolae or the newly formed t-tubule openings which allow entry of the MemBright to the tubule? Please clarify.

*Reviewer #3 (Recommendations for the authors):*

I commend the authors on an innovative and interesting investigation. I have made several suggestions in the section available for public view which I would encourage the authors to consider.

1. The pathological correlation is potentially of significant interest but is underdeveloped. In particular, I think the study would greatly benefit from a rigorous "rescue" experiment employing either experimentally manipulated murine or human skeletal muscle – for example with detubulation, genetic manipulation, or the myopathic samples you have. These seem like the ideal platform to test your putative mechanism to "regrow" t-tubules. These may get at the relative contribution of a Cav3-BIN1 mechanism versus other mechanisms active during early biogenesis. Is the CAV3-BIN 1 mechanism sufficient to rescue t-tubule pathology?

2. You should consider a simple screening western blot to map at least some of the other t-tubule regulating processes in your pathological and experimentally manipulated samples. Are JPH, Telethonin, and others constant? This may help understand whether there is any interplay between these pathways.

3. You mention in passing the observation of "polarized" caveolae rings – which are dense in one area – what is their significance? Have you observed a different fate for them compared to more symmetric rings?

4. What effect does unroofing have on membrane structure? Are you able to demonstrate that other ultrastructural features are unchanged?

5. You show that exon 11 is critical for tubule formation but not ring formation. Have you examined whether exon 11 alone is capable of ring formation, even in your supported lipid bilayer model? In an analogous fashion in heart, it was recently shown that JPH2- NT is capable of mediating most t-tubule regulating functions of Junctophilin. This may have consequences later on if these are conceived as gene therapies.

6. Your model scheme could include, at least in the legend, the dependence of tubule formation on exon 11.

7. You should contextualize your finding of exon 11 dependent tubule formation in the discussion. Indeed, the discussion does not adequately frame your findings in the context of either (i) what is already known about BIN 1 function and also (ii) how your findings fit into our overall understanding of multi-protein biogenesis of t-tubules. You should also mention possible translational relevance since you use a disease model.

8. A rescue experiment would involve combined or staged BIN1 and Cav3 overexpression with assays for t-tubule structure and function. This would determine whether the proposed mechanism is sufficient for t-tubule biogenesis. Likewise, more granular information on the typical temporospatial details of Cav3-BIN1 mediated tubulation is needed to fully understand their role in t-tubule biogenesis. Another understudied element is how t-tubule orientation is governed – as the authors point out immature tubules are generally oriented longitudinally. The live cell experiments shown may have promise in elucidating these features. Did you measure BIN1 protein expression in the experiments in Figures 7 and 8?

9. How does longitudinalization figure into your pathway in Figure 9? At what stage does this occur?

10. Your abstract mentions patients with BIN1 mutations but to my understanding, you did not include such patients, only those with Cav3 mutations.

11. Figure 8 – did you quantify the disordered ring structure in patient samples? You describe qualitatively but have you tried to objectively measure? This would support your statement on lines 326 and 327.

12. In your discussion you may consider commenting on the relevance to cardiac muscle.

13. Figure 9 illustration quality is poor, consider revising.

---

## [Author Response]

Reviewer #1 (Recommendations for the authors):The major issue that reduces the strength of several conclusions throughout the manuscript is the lack of quantification or appropriate controls for several imaging-based observations. Additional quantification throughout would add important context regarding the frequency, consistency, and effect size of key observations, and how they change with different molecular manipulations (such as Bin1 overexpression) or pathological variants. While the authors may have a very trained eye for detecting structural differences via EM, they cannot assume a broad readership will, and representative images are insufficient to support robust conclusions. The following 5 points are all related to this theme:1. The conclusion that Bin1GFP induced the formation of rings and tubules while the removal of exon 11 prevented tubule formation is insufficiently supported. Some quantification of ring and tubule prevalence should be provided.

We thank the reviewer for his suggestion. We performed new experiments to quantify the tubule and ring density between myotubes expressing the Bin1 isoform without exon11 (-ex11) and with exon 11 (+ex11). These experiments have now been added in a new Figure 5—figure supplement 1 which shows that both Bin1 isoforms are capable of producing rings but only isoform containing exon11 produces significant number of tubules. In addition, to strengthen the effect of Cav3 mutations on tubule density we quantified the ring and tubule density upon expression of Bin1+ex11 in myotubes from a patient with the R26Q mutation. These experiments have now been included in a new Figure 8 panel F, G and I and the text has been amended at lines 252-254.

2. For figure 7A-B, necks of caveolae are seen on the extracellular surface of intact cells overexpressing Bin1, but no similar EM of a non-Bin1 overexpressing cell or quantification is presented to show such structures do not exist or exist at lower frequency in the absence of Bin1 (or with Bin1exon11 mutant).

We agree with the reviewer that it would be good to show rings from the outside on intact myotubes. As expected, we have only found very few instances, one is shown in Author response image 1, but they are too rare to quantify as the criteria to decide whether it is a ring from the outside is subjective and therefore difficult to quantify. We are showing an example that we think corresponds to a caveolae ring as seen from the outside in control myotubes but we now clearly state this in the main text on line 300-301.

**Author response image 1. sa2fig1:** 

3. For Figure 8A-C, it is claimed that the EM images show altered structure and spatial organization of caveolae rings with rare supernumerary caveolae in R26Q variants. This is not clear nor particularly interpretable without some sort of quantification of these morphological changes and their occurrence. It is not clear whether the reduction in Cav3 protein results in altered rings, or simply fewer rings, and this is an important distinction for understanding pathomechanisms.

We agree with reviewer#1 that answering this point would considerably strengthen our conclusions on caveolinopathy patients. In order to substantiate our morphological PREM data, we have now quantified the caveolae ring diameter and density formed at the surface of patient myotubes. We also quantified the density of tubules present in patient myotubes expressing Bin1 compared to con-trol myotubes. These results have been included in Figure 8 panel F, G and I. We found that there are significantly less caveolae rings at the surface of patient myotubes (in agreement with a de-creased Cav3 surface expression) and their average diameter is increased compared to control myotubes. Following reviewer #3’s advice, we successfully rescued this phenotype by producing a patient myotube line re-expressing full-length Cav3. We were able to rescue the ring density while reducing the average diameter to control values. These results have now been added in a new Figure 8.

4. For Figure 8H-K, there must also be some quantification of the observed triad abnormalities, and again a comparison to control samples, to have this be a meaningful contribution. Taken together with the prior point, this limits any insight into pathology, and the claim in discussion (line 436) that caveolae ring formation is defective in patient myotubes is insufficiently supported.

We agree with reviewer #1 that a comparison to control triads should be shown. We have now included an EM image representing a control triad. However, both T-tubule swelling and T-tubule bifurcations are rare events in caveolinopathy patients and cannot be quantified easily for any meaningful point to emerge. We have now clearly stated this point in the Results section on lines 355-357.

5. The addition of actin filaments in the model – particularly in the early stages of ring formation- seems speculative and potentially misleading. The authors' in vitro data in lipid bilayers (that presumably lack actin filaments) show that actin is dispensable for ring formation and even early tube formation, and actin is not mentioned in the main text. I would advise removing it from the cartoon.

Although actin filaments are very abundant around caveolae rings on our PREM micrographs, we agree with the reviewer that adding them on the model is too speculative. We removed them from the improved version of Figure 9 cartoon.

Reviewer #2 (Recommendations for the authors):This manuscript is very interesting and the data are compelling. I do however have some additional comments below for the authors' consideration.The authors should clarify the statistics used. They state that 'n' is the number of independent biological replicates, but it is not always clear what the biological replicate is. For example, in Figure 1B is 'n' the ring or the myotube, in Fig1J is it the caveolae or the experimental repeat (which I think it should be)? It would be useful for this clarity throughout.

We thank reviewer #2 for deeming our data compelling. We have now clarified the statistics used in the appropriate figure legends throughout the manuscript and clearly define biological or technical replicates.

In Figure 5G do the conditions or the Bin1 isoform vary between the upper and lower panels? Can this be added to the legend? It is not quite clear to me what is being shown in Figure 6E.

In Figure 5G, the lower and upper panel both show exactly the same conditions. We have now clarified the panel 5G of new Figure 5. We have also removed Figure 6E for clarity as it didn’t provide any essential information.

The legend for Figure 6B-E is identical to 6F – could they be combined to simplify? Similarly for Figures6G and H?

As suggested by reviewer #2, we have simplified the legends for Fig. 6.

Lines 288-289 suggest BIN1 is not always localised to caveolae but the authors might consider if it would be more convincing for the data cited to include cav3 labelling.

As suggested by reviewer #2, we have simplified the legends for Figure 6.

In Fig7A are the authors suggesting the openings are caveolae or the newly formed t-tubule openings which allow entry of the MemBright to the tubule? Please clarify.

We apologize for the wording we chose. We meant that Bin1 is not always localized to caveolae rings as caveolae ultrastructural characteristics are absent. CLEM would not a convincing technique to demonstrate the absence of caveolin on these structures. We have clarified the text on line 283.

Reviewer #3 (Recommendations for the authors):I commend the authors on an innovative and interesting investigation. I have made several suggestions in the section available for public view which I would encourage the authors to consider.1. The pathological correlation is potentially of significant interest but is underdeveloped. In particular, I think the study would greatly benefit from a rigorous "rescue" experiment employing either experimentally manipulated murine or human skeletal muscle – for example with detubulation, genetic manipulation, or the myopathic samples you have. These seem like the ideal platform to test your putative mechanism to "regrow" t-tubules. These may get at the relative contribution of a Cav3-BIN1 mechanism versus other mechanisms active during early biogenesis. Is the CAV3-BIN 1 mechanism sufficient to rescue t-tubule pathology?

We agree with reviewer #3 that a rescue experiment could be ideal to convincingly demonstrate our claims. While an attempt to regrow tubules is not feasible with our current tools, we performed a rescue experiment aiming at re-expressing Cav3 in the patient myotubes by using a lentivirus and correct the two defective parameters: caveolae ring density and caveolae ring diameter. The results are now shown in Figure 8 F and G.

2. You should consider a simple screening western blot to map at least some of the other t-tubule regulating processes in your pathological and experimentally manipulated samples. Are JPH, Telethonin, and others constant? This may help understand whether there is any interplay between these pathways.

In agreement with reviewer #3’s point we performed a screening western blot experiments to compare protein markers from the different compartments involved. We used antibodies against DHPR a1s-subunit and Bin1 as markers for the t-tubule, RyR1 as a *bona fide* marker for sarcoplasmic reticulum terminal cisternae and Junctophilin 2 (JPH2) as a marker for both sides of the triad. Our experiments using two patient cell lines show no difference in expression levels for these markers. These results are now shown in Figure 8—figure supplement 1.

3. You mention in passing the observation of "polarized" caveolae rings – which are dense in one area – what is their significance? Have you observed a different fate for them compared to more symmetric rings?

This is a very interesting point. Indeed, we often observe an accumulation of caveolae on the ring which are dense in one area. Our time-lapse experiments with Cav3-GFP show a very bright spot prior to seeing a tubule extend from the ring suggesting that in order to grow a tubule, caveolae need to accumulate at a specific site on the ring perimeter. We have now included at lines 231-233 in the text to state that “Interestingly, Cav3 signal increased at the site from which the tubule emanated (yellow arrows in Figure 4, A and B) suggesting that the polarized accumulation of caveolae could precede tubulation” and also added yellow arrows in Figure 4 to point out the Cav3 accumulation prior to the tubule emanation.

**Author response image 2. sa2fig2:** Time-lapse imaging of Cav3/Bin1 tubules extending from rings. Gallery of consecutive frames from the time-lapse sequences of Cav3GFP expressing myotubes from figure 4 and movies 5 and 6. The gallery shows consecutive frames every 10s of a tubule emanating from a ring (see Video 5 and Video 6). Note the bright spot on the region of the ring from which the tubule emanates.

4. What effect does unroofing have on membrane structure? Are you able to demonstrate that other ultrastructural features are unchanged?

The unroofing step is performed here by applying brief sonicator pulses on the cells at the lowest delivered power setting. Depending on the degree of unroofing some cells will be completely opened and others less. As with other EM techniques, unroofing, fixation, subsequent drying and substituting can all produce artefacts. However, we carefully check on our replicas that other ultra-structural features including the membrane itself, ER tubules, clathrin-coated pits and plaques, caveolae, actin filaments, microtubules and intermediate filaments are presenting a normal ultra-structural morphology.

5. You show that exon 11 is critical for tubule formation but not ring formation. Have you examined whether exon 11 alone is capable of ring formation, even in your supported lipid bilayer model? In an analogous fashion in heart, it was recently shown that JPH2- NT is capable of mediating most t-tubule regulating functions of Junctophilin. This may have consequences later on if these are conceived as gene therapies.

Our results show that exon 11 is not required for ring formation as both Bin1 isoforms form them. In our experiments exon11 is important for tubule formation and elongation from the rings. It would be interesting in future experiments to test the behavior of exon 11 alone in an in vitro setting on supported lipid bilayers.

6. Your model scheme could include, at least in the legend, the dependence of tubule formation on exon 11.

We agree with this point raised by reviewer 3 and have now amended our cartoon to show the dependence of tubule formation on exon 11.

7. You should contextualize your finding of exon 11 dependent tubule formation in the discussion. Indeed, the discussion does not adequately frame your findings in the context of either (i) what is already known about BIN 1 function and also (ii) how your findings fit into our overall understanding of multi-protein biogenesis of t-tubules. You should also mention possible translational relevance since you use a disease model.

We agree with this reviewer and this comment was also raised by reviewer#1. We have now quantified the effect of +/- exon11 on ring density and tubule density. We show that exon11 is not required for ring formation as both Bin1 isoforms are capable of forming them at the myotube surface while only Bin1+ex11 is capable of tubulating the membrane. These results have indeed important translational implications as they identify an important defect that could serve as a read-out to develop therapies.

8. A rescue experiment would involve combined or staged BIN1 and Cav3 overexpression with assays for t-tubule structure and function. This would determine whether the proposed mechanism is sufficient for t-tubule biogenesis. Likewise, more granular information on the typical temporospatial details of Cav3-BIN1 mediated tubulation is needed to fully understand their role in t-tubule biogenesis. Another understudied element is how t-tubule orientation is governed – as the authors point out immature tubules are generally oriented longitudinally. The live cell experiments shown may have promise in elucidating these features. Did you measure BIN1 protein expression in the experiments in Figures 7 and 8?

To follow the reviewer’s recommendation, we included in the revised manuscript a rescue experiment by Cav3 overexpression and several of our movies specify the spatio-temporal details of T-tubule formation. We agree with reviewer #3 that understanding T-tubule orientation is a particularly relevant question. Indeed, time-lapse imaging shows that T-tubules very often extend in the longitudinal direction. Although the experiments aiming at understanding how is the tubule oriented during its formation extends the purpose of this work. We have now performed Western blot analysis of Bin1 protein levels in control vs caveolinopathy patient myotubes (two patients with the R26Q mutation). These results show no change in Bin1 expression and are included in a new Figure 8—figure supplement 1.

9. How does longitudinalization figure into your pathway in Figure 9? At what stage does this occur?

It has been systematically reported in the literature that during development and in vitro, T-tubules grow longitudinally until post-natal days 6 to 10 in mice where tubules adopt a transversal orientation (Franzini-Armstrong C. 1991. *Dev Biol)*. We also observe this, and we have hypothesized that the actin stress fibers or cortical microtubules might contribute to this longitudinal extension. When the sarcomeric apparatus has matured enough, after birth and formation of the NMJ is established, a new wave of transversal tubes might extend to form the adult T-system. However, this particular aspect of T-tubule formation and maturation is not addressed by this work focused on initial formation steps.

Franzini-Armstrong C. Simultaneous maturation of transverse tubules and sarcoplasmic reticulum during muscle differentiation in the mouse. Dev Biol. 1991 Aug;146(2):353-63. doi: 10.1016/0012-1606(91)90237-w. PMID: 1864461.

10. Your abstract mentions patients with BIN1 mutations but to my understanding, you did not include such patients, only those with Cav3 mutations.

We agree with reviewer #3. We have now added the word " caused by Cav3 or Bin1 dysfunction".

11. Figure 8 – did you quantify the disordered ring structure in patient samples? You describe qualitatively but have you tried to objectively measure? This would support your statement on lines 326 and 327.

We now included the quantification of ring density and diameter in control vs patient myotubes. We observed that in patient myotubes, there is a significant decrease in ring density, associated with the reduced Cav3 levels at the surface. We also found that in agreement with our PREM data, the average diameter is increased in caveolinopathy patient cells. By rescuing Cav3 expression in the patient myotubes, we rescued the density and reduced the average ring diameter.

12. In your discussion you may consider commenting on the relevance to cardiac muscle.

We have modified the text to add this point: "Importantly, these mechanisms could also contribute to biogenesis of T-tubules in the heart as Cav3, cavin 1 and cavin 4 mutations lead to cardiomyopathies (Rajab et al., 2010; Rodriguez et al., 2011; Vatta et al., 2006)."

13. Figure 9 illustration quality is poor, consider revising.

We have now improved Figure 9 cartoon by removing actin filaments as suggested by reviewer #1, and have included additional text and color to describe the difference between Bin1 with or without exon 11 in neither ring or tubule formation.